# MemEvolve: Meta-Evolution of Agent Memory Systems

**Guibin Zhang** [* 1]   **Haotian Ren** [* 2]   **Chong Zhan** [2]   **Junhao Wang** [3]   **He Zhu** [4]   **Wangchunshu Zhou** [5]
**Shuicheng Yan** [1]

## Abstract

Self-evolving memory systems are rapidly reshaping the evolutionary paradigm of large language model (LLM)-based agents. Prior work has predominantly relied on manually engineered memory architectures to store trajectories, distill experience, and synthesize reusable tools, enabling agents to evolve on the fly within environment interactions. However, this paradigm is fundamentally constrained by the *staticity* of the memory system itself: while memory facilitates agent-level evolving, the underlying memory architecture cannot be meta-adapted to diverse task contexts. To address this gap, we propose `MemEvolve`, a meta-evolutionary framework that jointly evolves agents' experiential knowledge and their memory architecture, allowing agent systems not only to accumulate experience but also to progressively refine how they learn from it. To ground `MemEvolve` in prior work and promote openness in future self-evolving systems, we introduce `EvolveLab`, a unified memory codebase that distills twelve representative memory systems into a modular design space (*encode*, *store*, *retrieve*, *manage*), providing a standardized implementation substrate and a fair experimental arena. Extensive evaluations on four challenging agentic benchmarks show that `MemEvolve` delivers (i) substantial performance gains, improving frameworks such as SmolAgent and Flash-Searcher by up to 17.06%, and (ii) strong cross-task and cross-LLM generalization, yielding memory architectures that transfer effectively across diverse benchmarks and backbones. Codes are available at here.

[1]National University of Singapore [2]Beijing University of Posts and Telecommunications [3]Chinese University of Hong Kong [4]OPPO [5]Bytedance Inc. . Correspondence to: Wangchunshu Zhou <chunshu@bytedance.com>, Shuicheng Yan <yansc@nus.edu.sg>.

*Proceedings of the 43rd International Conference on Machine Learning*, Seoul, South Korea. PMLR 306, 2026. Copyright 2026 by the author(s).

## 1. Introduction

Language agents and agent systems, empowered by increasingly capable foundation models (Team et al., 2025a;b) and sophisticated scaffolding (Wang et al., 2024a; LangChain, 2023), have advanced rapidly, demonstrating unprecedented performance across complex tasks such as deep research (Chen et al., 2025), scientific discovery (Bai et al., 2025; Wei et al., 2025b), and industrial report generation (Zhang et al., 2025g). A key driving force behind this success is the *agent memory system* (Zhang et al., 2024b; Hu et al., 2025c), which persistently captures interactions between the agent and environment, distilling them into diverse forms of knowledge and skills, and thereby enabling large language model (LLM)-based agents to evolve continuously in task solving and world exploration.

Naturally, the choice of memory paradigm plays a decisive role in shaping an agent's capacity for on-the-fly self-evolution. Initial designs centered on raw trajectory storage and few-shot prompting (Zhong et al., 2024; Wen et al., 2024), which were later superseded by more abstracted textual artifacts such as tips, shortcuts, and reasoning templates (Ouyang et al., 2025; Zhang et al., 2025b; Ye et al., 2025; Tang et al., 2025). Recent advances have also explored structured tool interfaces (*e.g.*, APIs (Zheng et al., 2025), MCPs (Qiu et al., 2025b;a; Zhang et al., 2025h)) and code-level repositories (Zhang et al., 2025e; Wang et al., 2025a) as memory carriers. Amid this growing diversity, an inquisitive practitioner might ask: *What kind of memory architecture most effectively drives agent self-improving?*

We posit that no universally optimal memory architecture exists. Memory systems tailored to distill reusable APIs from trajectories may excel in web-centric tasks yet offer limited benefit for mathematical or scientific reasoning, while self-critique–based memories, though effective in reasoning-intensive domains (Cai et al., 2025), often underperform in coding and tool-use settings (Zhang et al., 2025d). These trade-offs stem from the *static nature* of current memory pipelines, which are typically fixed in their ingestion, abstraction, and retrieval design (Zhang et al., 2025i). By contrast, effective human learners are capable of adapting their learning strategies to the task, *e.g.*, prioritizing memorization for literary analysis while abstracting solution templates

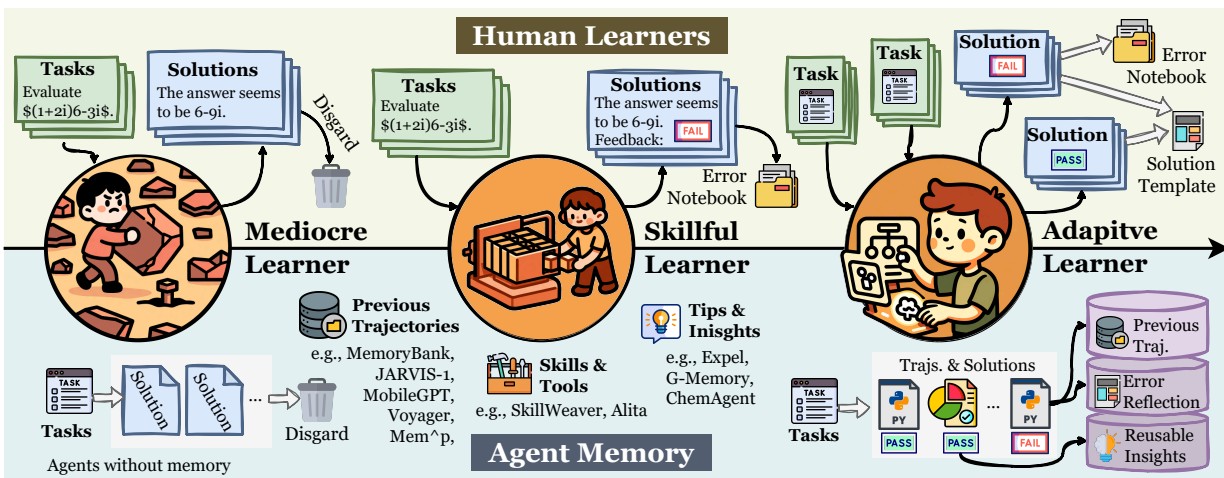

*Figure 1.* The paradigm of agent self-evolution admits a natural analogy to human learning. At one extreme, a *mediocre learner* fails to benefit from experience (agents without memory). More capable *skillful learners* can extract reusable skills from past experience, albeit through a pre-defined abstraction scheme. In contrast, an *adaptive learner* simultaneously accumulates experience and dynamically adjusts the strategy by which experience is consolidated and utilized. This final regime precisely characterizes the objective of `MemEvolve`.

for mathematics. We argue that agent memory systems must similarly transition from merely *skillful* to *adaptive*, enabling the memory architecture itself to *meta-evolve* in response to task demands while preserving generalizability, as shown in Figure 1. To put it more formally:

> How can a memory system not only facilitate the agent system's evolution but also *meta-evolve its own architecture* to achieve superior task-domain performance gains while preserving generalizability?

To address the challenge, we introduce `MemEvolve`, a framework that facilitates the dual evolution of an agent's experience and its memory architecture. Conceptually, `MemEvolve` operates as a bilevel optimization process: the inner loop performs a *first-order evolution*, where the agent, guided by a fixed memory system, adapts to a continuous stream of new tasks by populating its experience base. The outer loop drives a *second-order evolution*, meta-learning a more effective memory architecture to accelerate future learning. This allows the agent not only to evolve, but to evolve more efficiently and intelligently over time.

However, the vast and heterogeneous memory design space (*e.g.*, knowledge graphs, skill libraries, vector databases) poses a major challenge for controllable optimization. To make this tractable, we introduce a modular decomposition of memory architectures into four components: ♣ *Encode* (experience perception and formatting), ♦ *Store* (information commitment), ♥ *Retrieve* (context-aware recall), and ♠ *Manage* (consolidation and forgetting). `MemEvolve` evolves the programmatic implementations of these modules in a model-driven manner using inner-loop performance feedback, forming a virtuous cycle in which improved architectures enhance agent learning, and stronger agents yield

higher-quality trajectories that sharpen the outer-loop fitness signal for subsequent evolution.

To ground our framework within the diverse landscape of existing self-improving agent memories, we systematically re-implement twelve representative architectures in a unified modular design space, including EXPEL (Zhao et al., 2024), AGENT WORKFLOW MEMORY (Wang et al., 2024b), and DYNAMIC CHEATSHEET (Suzgun et al., 2025). The resulting framework, denoted as `EvolveLab`, serves both as an empirical foundation for `MemEvolve`'s evolutionary process and as a standardized codebase to facilitate future self-evolving research. Our contributions are as follows:

❶ **Unified Codebase:** We introduce `EvolveLab`, a modular design space for self-improving agent memory systems encompassing four key components (*encoding*, *storage*, *retrieval*, and *management*), providing unified implementations and benchmark support for a wide range of prevailing agent memory systems.

❷ **Meta-Evolution Framework:** We propose `MemEvolve`, a meta-evolutionary framework that jointly evolves both agents' experiential knowledge and their underlying memory architecture, in which agent systems not only accumulate experience but also progressively refine their mechanism for learning from it.

❸ **Experimental Evaluation:** Extensive experiments on four challenging agentic benchmarks demonstrate that `MemEvolve` delivers (I) **substantial performance gains**, improving frameworks such as SmolAgent and Flash-Searcher by up to $17.06\%$; and (II) **cross-domain, cross-framework and cross-LLM generalization**, where memory systems evolved on TaskCraft yield $2.0-9.09\%$ gains with unseen benchmarks and backbone models.

## 2. Related Work

**LLM Agent Systems.** Recent work has rapidly advanced LLM-based agent systems along two axes (Tran et al., 2025; Fang et al., 2025a). In terms of **system complexity**, research has evolved from single-agent workflows with manual tool orchestration (Wu et al., 2023; Significant-Gravitas, 2023) to multi-agent frameworks with automated coordination and rich MCP integration (Zhang et al., 2024a; 2025a; Wang et al., 2025b; Zhang et al., 2025c). Along the **task domain** axis, capabilities have expanded from coding and mathematical reasoning (Hong et al., 2024; Yin et al., 2023) to deep research and scientific discovery (Du et al., 2025; Ghareeb et al., 2025). A growing set of open-source systems now achieve competitive performance on challenging benchmarks such as (Mialon et al., 2023; Phan et al., 2025; Wei et al., 2025a; Chen et al., 2025), including OWL (Hu et al., 2025a), CK-PRO (Fang et al., 2025c), AGENTORCHES-TRA (Zhang et al., 2025f), and AIME (Shi et al., 2025b).

**Agent Memory Architectures.** Agent memory systems are commonly categorized into *personalized* and *self-improving* memories (Zhang et al., 2024b; Hu et al., 2025c), where the former captures user-specific preferences and the latter distills knowledge and skills from continual interaction, which is the focus of this work. Early self-improving systems stored raw trajectories as few-shot exemplars (Wang et al., 2023; Zhong et al., 2024; Packer et al., 2023), while later approaches abstracted experience into higher-level insights, procedural guidance, and reusable tools or structured repositories (Yang et al., 2025; Sun & Zeng, 2025; Wu et al., 2025b; Wang et al., 2025c; Zheng et al., 2025; Fang et al., 2025b; Zhao et al., 2025; Qiu et al., 2025a;b; Zhang et al., 2025e). Despite representational differences, these methods share a common goal: enabling agents to learn, adapt, and improve through continual experience.

## 3. EvolveLab Codebase

In this section, we first formalize the LLM agentic system and its associated memory architecture, then present the modular design space of `EvolveLab`, and finally introduce the unified codebase `EvolveLab`.

### 3.1. Preliminary

We formalize an LLM-based agentic system as $\mathcal{M} = \langle \mathcal{I}, \mathcal{S}, \mathcal{A}, \Psi, \Omega \rangle$, where $\mathcal{I}$ indexes the $\{1, \cdots, N\}$ agents, $\mathcal{S}$ denotes the shared state space, $\mathcal{A} = \bigcup_{i \in \mathcal{I}} \mathcal{A}_i$ represents the joint action space, and $\Psi(s_{t+1} \mid s_t, a_t, \mu(t))$ describes the environment dynamics with $\mu(t) \in \mathcal{I}$ indicating the active agent at time step $t$. The system leverages a memory module $\Omega$, which maintains a continuously evolving memory state $M_t$. At each step, the active agent observes the current state $s_t$, considers a task-specific query $\mathcal{Q}$, and interacts with $\Omega$ to retrieve contextually relevant memory

$c_t$, conditioned on its interaction history $\mathcal{H}_t$. The agent $\mu_t$'s policy $\pi_{\mu_t}$ then delivers an action:

$$a_t = \pi_{\mu(t)}(s_t, \mathcal{H}_t, \mathcal{Q}, c_t), \ c_t \sim \Omega(M_t, s_t, \mathcal{H}_t, \mathcal{Q}). \quad (1)$$

Following task execution, a trajectory $\tau = (s_0, a_0, \ldots, s_T)$ is recorded, with an overall performance evaluated via a terminal reward $R(\tau)$. The memory system assimilates new experience units $\epsilon$, which can vary in granularity (from individual state-action transitions to aggregated segments or complete trajectories), and updates the memory state as $M_{t+1} = \Omega(M_t, \epsilon)$, where $\Omega$ abstracts the memory's mechanisms for integrating and organizing new knowledge.

### 3.2. Modular Design Space of Memory Systems

The heterogeneous and rapidly evolving landscape of self-improving agent memories presents challenges for systematic analysis and controlled experimentation. To address this, we propose a modular design space that decomposes any memory system $\Omega$ into four functionally distinct yet interdependent components: $\Omega = (\mathcal{E}, \mathcal{U}, \mathcal{R}, \mathcal{G})$, representing *encode*, *store*, *retrieve*, and *manage* operations, respectively. ♣ **Encode** ($\mathcal{E}$): Transforms raw experiences, such as trajectory segments $\tau_t = (s_t, a_t, s_{t+1})$, tool outputs, or self-critiques, into structured representations $e_t = \mathcal{E}(\epsilon_t)$. Encoding may be as simple as compressing raw traces (Zheng et al., 2023) or as sophisticated as extracting generalizable lessons (Zheng et al., 2025). ♦ **Store** ($\mathcal{U}$): Integrates encoded experiences into the persistent memory $M_t$, yielding $M_{t+1} = \mathcal{U}(M_t, e_t)$. Storage can be vector databases (Zhao et al., 2024), knowledge graphs (Zhang et al., 2025b; Rasmussen et al., 2025), or others. ♥ **Retrieve** ($\mathcal{R}$): Provides task-relevant memory content, formalized as $c_t = \mathcal{R}(M_t, s_t, \mathcal{Q})$, which informs the agent's policy decision $a_t$. Retrieved content may include reusable tools (Zhang et al., 2025f), planning experience (Tang et al., 2025), or distilled procedural knowledge (Wu et al., 2025b; Yang et al., 2025; Fang et al., 2025b). ♠ **Manage** ($\mathcal{G}$): Performs offline and asynchronous operations such as consolidation, abstraction, or selective forgetting to maintain long-term memory quality and efficiency, denoted as $M_t' = \mathcal{G}(M_t)$.

This modular abstraction allows us to represent each memory system as a specific combination of programmatic implementations for $(\mathcal{E}, \mathcal{U}, \mathcal{R}, \mathcal{G})$, forming a "genotype" that facilitates the meta-evolutionary process of `MemEvolve`.

### 3.3. EvolveLab Codebase

Based on the above design space, we introduce `EvolveLab`, a unified and extensible codebase designed for the systematic implementation and evaluation of self-evolving memories, serving as a standardized resource for the community.

*Table 1.* Self-improving memory systems implemented in `EvolveLab`. In "Mul." column, 🧍 indicates support for single-agent settings, while 🧑‍🤝‍🧑 denotes compatibility with multi-agent systems. "Gran." specifies the granularity at which memory is provided (*step-wise* vs. *trajectory-wise*), and "Online" indicates whether memory is updated *on-the-fly* (🔗) or maintained as an offline repository (🔁).

| Method | Date | Mul. | Gran. | Online | Encode | Store | Retrieve | Manage |
|---|---|---|---|---|---|---|---|---|
| I. Voyager | 2023.5 | 🧍 | traj. | 🔗 | Traj. & Tips | Vector DB | Semantic Search | N/A |
| II. ExpeL | 2023.8 | 🧍 | traj. | 🔗 | Traj. & Insights | Vector DB | Contrastive Comparison | N/A |
| III. Generative | 2023.10 | 🧑‍🤝‍🧑 | traj. | 🔗 | Traj. & Insights | Vector DB | Semantic Search | N/A |
| IV. DILU | 2024.2 | 🧍 | traj. | 🔗 | Traj. | Vector DB | Semantic Search | N/A |
| V. AWM | 2024.9 | 🧍 | traj. | 🔗🔁 | Workflows | Vector DB | Semantic Search | N/A |
| VI. Mobile-E | 2025.1 | 🧍 | step | 🔁 | Tips & Shortcuts | Vector DB | Semantic Search | N/A |
| VII. Cheatsheet | 2025.4 | 🧍 | traj. | 🔗 | Tips & Shortcuts | JSON | Semantic Search | N/A |
| VIII. SkillWeaver | 2025.4 | 🧍 | traj. | 🔁 | APIs | Tool Library | Function Matching | Skill Pruning |
| IX. G-Memory | 2025.6 | 🧑‍🤝‍🧑 | traj. | 🔗 | Tips & Workflow | Graph | Graph/Semantic Search | Episodic Consolidation |
| X. Agent-KB | 2025.7 | 🧑‍🤝‍🧑 | step | 🔁 | Tips & Workflow | Hybrid DB | Hybrid Search | Deduplication |
| XI. Memp | 2025.8 | 🧍 | step. | 🔗 | Tips & Workflow | JSON | Semantic Search | Failure-driven Adjustment |
| XII. EvolveR | 2025.10 | 🧍 | step. | 🔗 | Tips & Workflow | JSON | Contrastive Comparison | Update & Pruning |

**Implementation.** The cornerstone of `EvolveLab` is its modular and hierarchical design. Every memory architecture re-implemented in our codebase (see Table 1) inherits from a singular abstract base class, BASEMEMORYPROVIDER, which enforces the unified four-component interface: ♣ Encode, ♦ Store, ♥ Retrieve, and ♠ Manage. This ensures that diverse memory mechanisms can be managed, modified, and evolved under a consistent programmatic structure. The framework offers out-of-the-box support for multiple challenging benchmarks in both *online* and *offline* manner. More details are at Section A.

## 4. MemEvolve: Meta-Evolving Memory

### 4.1. Dual-Evolution Process

Traditional self-improving memory systems operate under a *fixed memory architecture*, where the memory interface $\Omega$ is predefined and remains static. Within this architecture, the agent iteratively populates and updates its memory state $M_t$ through interaction with the environment and task experiences. For a trajectory $\tau$ induced by a query $\mathcal{Q}$, the memory evolution follows

$$M_{t+1} = \Omega(M_t, \epsilon_\tau), \quad \epsilon_\tau \in \mathcal{E}(\tau), \quad (2)$$

where $\mathcal{E}(\cdot)$ denotes an experience extraction operator that maps a trajectory to a set of experience units, and $\epsilon_\tau$ is an element sampled from this set. While this enables the accumulation of knowledge, it fundamentally precludes *architectural adaptation*, as the memory interface $\Omega$ itself remains immutable.

To transcend this limitation, we propose a **dual-evolution process** that jointly evolves (i) the agent's memory base and (ii) the underlying memory architectures (as illustrated in Figure 2). Instead of a single static $\Omega$, we maintain, at each evolutionary iteration $k$, a finite set of candidate memory systems $\{\Omega_j^{(k)}\}_{j \in \mathcal{J}^{(k)}}$, where each $\Omega_j^{(k)}$ is instantiated as a concrete realization of the four-component memory interface $\Omega_j^{(k)} \triangleq (\mathcal{E}_j^{(k)}, \mathcal{U}_j^{(k)}, \mathcal{R}_j^{(k)}, \mathcal{G}_j^{(k)})$. The initial iteration start from a singleton set $|\mathcal{J}^{(0)}| = 1$, corresponding to a

hand-designed baseline memory, while later iterations admit multiple competing candidates. Given a batch of trajectories $\mathcal{T}_j^{(k)}$ independently generated by executing the agent with memory system $\Omega_j^{(k)}$, the dual-evolution process consists of two nested loops:

- **Inner Loop (Experience Evolution).** For each candidate memory system $\Omega_j^{(k)}$, the associated memory state $M_{t,j}^{(k)}$, initialized as an empty memory at the beginning of iteration $k$, is updated along trajectories $\tau \in \mathcal{T}_j^{(k)}$ via

$$M_{t+1,j}^{(k)} = \Omega_j^{(k)}\big(M_{t,j}^{(k)}, \epsilon_\tau\big), \quad \epsilon_\tau \in \mathcal{E}_j^{(k)}(\tau). \quad (3)$$

Executing the agent with $\Omega_j^{(k)}$ over $\mathcal{T}_j^{(k)}$ yields, for each trajectory $\tau$, a feedback vector $\mathbf{f}_j(\tau) \in \mathbb{R}^d$, where $d = 3$ corresponds to the number of evaluation metrics (*i.e.*, task success, token consumption, and latency). An aggregation operator $\mathcal{S}$ summarizes the inner-loop outcomes for each candidate as

$$\mathbf{F}_j^{(k)} = \mathcal{S}\big(\{\mathbf{f}_j(\tau)\}_{\tau \in \mathcal{T}_j^{(k)}}\big), \quad j \in \mathcal{J}^{(k)}. \quad (4)$$

- **Outer Loop (Architectural Evolution).** The set of memory architectures is then updated based on the collection of summaries $\{\mathbf{F}_j^{(k)}\}_{j \in \mathcal{J}^{(k)}}$. A meta-evolution operator $\mathcal{F}$ selects high-performing candidates and proposes new variants, producing the next iteration's candidate set:

$$\{\Omega_{j'}^{(k+1)}\}_{j' \in \mathcal{J}^{(k+1)}} = \mathcal{F}\Big(\{\Omega_j^{(k)}\}_{j \in \mathcal{J}^{(k)}}, \{\mathbf{F}_j^{(k)}\}_{j \in \mathcal{J}^{(k)}}\Big). \quad (5)$$

Specifically, $\mathcal{F}$ ranks candidates according to $\mathbf{F}_j^{(k)}$, retains the top-$K$ memory systems, and generates new architectures by modifying or recombining all four components $(\mathcal{E}, \mathcal{U}, \mathcal{R}, \mathcal{G})$ of the selected candidates, where $K$ denotes a fixed survivor budget. We detail the implementation of $\mathcal{F}(\cdot)$ in Section 4.2.

**Unified view.** At a higher level, each iteration $k$ alternates between (i) evolving the *memory experience base* from an

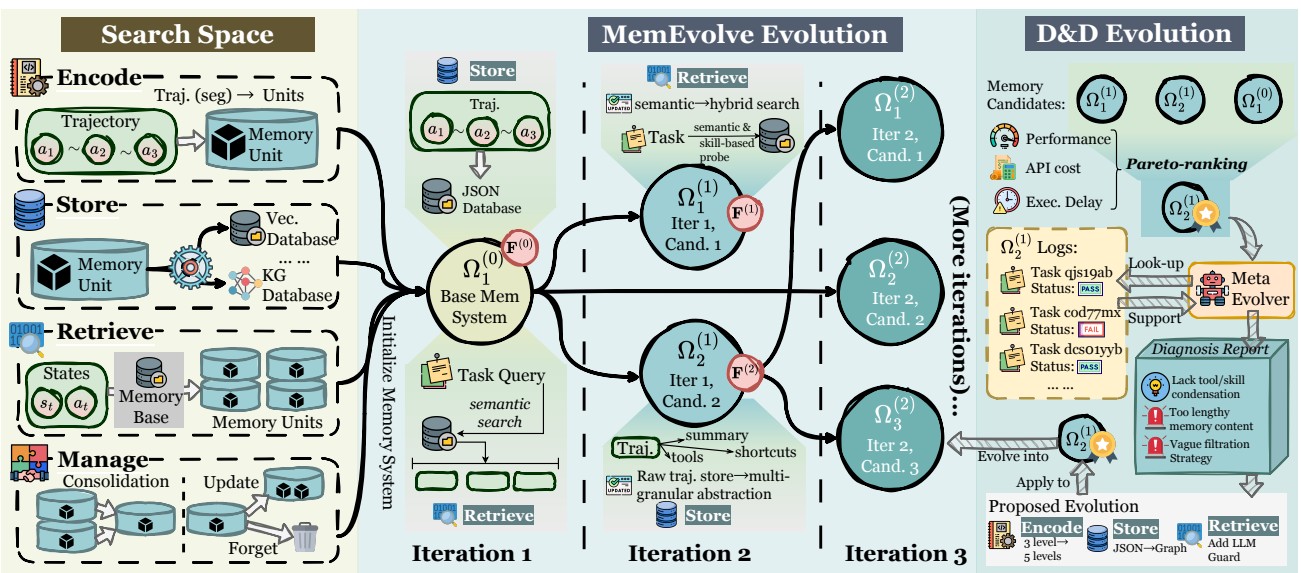

*Figure 2.* The overview meta-evolving process of our proposed `MemEvolve`.

empty initialization under a fixed set of architectures, and (ii) evolving the *memory architectures* themselves based on the induced performance:

$$
\begin{aligned}
\big(\{\varnothing\}_{j\in\mathcal{J}^{(k)}}, \{\Omega_j^{(k)}\}_{j\in\mathcal{J}^{(k)}}\big) &\xrightarrow{\text{inner}} \big(\{M_{t+1,j}^{(k)}\}_j, \{\Omega_j^{(k)}\}_j\big) \\
&\xrightarrow{\text{outer}} \big(\{M_{t+1,j}^{(k)}\}_j, \{\Omega_{j'}^{(k+1)}\}_{j'\in\mathcal{J}^{(k+1)}}\big).
\end{aligned}
$$
(6)

By iterating this dual-evolution process, the agent does not merely accumulate experience within a fixed memory system; instead, both the memory base and the governing memory architectures co-evolve, yielding increasingly adaptive and resource-aware memory-driven behavior over time.

### 4.2. Diagnose-and-Design Evolution

We now detail the meta-evolution operator $\mathcal{F}$, which governs the architectural update in each evolutionary iteration. Conceptually, $\mathcal{F}$ decomposes into two coordinated components: (i) *architectural selection*, which identifies a subset of high-performing memory systems to serve as evolutionary parents, and (ii) *diagnose-and-design evolution*, which generates new memory architectures from each selected parent through a structured diagnosis procedure followed by a constrained redesign within the modular space.

**Architectural Selection.** Given the candidate set $\{\Omega_j^{(k)}\}_{j\in\mathcal{J}^{(k)}}$ and their corresponding summaries $\{\mathbf{F}_j^{(k)}\}$, we define each summary vector as

$$
\mathbf{F}_j^{(k)} \triangleq \big(\text{Perf}_j^{(k)}, -\text{Cost}_j^{(k)}, -\text{Delay}_j^{(k)}\big),
$$
(7)

where higher values are preferred in all dimensions. Candidates are first ranked by non-dominated sorting over $\mathbf{F}_j^{(k)}$, yielding a Pareto rank $\rho_j^{(k)}$. Within the same Pareto rank, candidates are further ordered by performance $\text{Perf}_j^{(k)}$. The

top-$K$ candidates are selected as the parent set:

$$
\mathcal{P}^{(k)} = \underset{j\in\mathcal{J}^{(k)}}{\text{Top-K}}\big(\rho_j^{(k)},\ \text{Perf}_j^{(k)}\big).
$$
(8)

This selection step ensures that architectural evolution is guided by systems that exhibit favorable trade-offs between task effectiveness and resource efficiency, while prioritizing task performance among Pareto-equivalent candidates.

**Diagnose-and-Design Evolution.** For each parent architecture $\Omega_p^{(k)} \in \mathcal{P}^{(k)}$, $\mathcal{F}$ generates a set of $S$ descendants $\{\Omega_{p,s}^{(k+1)}\}_{s=1}^S$ through a two-phase process:

- **Diagnosis.** Each parent architecture is examined using trajectory-level evidence from its own execution batch $\mathcal{T}_p^{(k)}$. For each trajectory, the agent provides outcome statistics (*e.g.*, success indicators, token costs) together with a structured description of the associated task query. A replay interface grants access to the corresponding trajectories $\tau \in \mathcal{T}_p^{(k)}$, enabling targeted inspection of memory behavior, including retrieval failures, ineffective abstractions, or storage inefficiencies. The diagnosis phase thus produces a structured defect profile $\mathcal{D}(\Omega_p^{(k)})$, characterizing architectural bottlenecks across the four memory components $\big(\mathcal{E}_p^{(k)}, \mathcal{U}_p^{(k)}, \mathcal{R}_p^{(k)}, \mathcal{G}_p^{(k)}\big)$.

- **Design.** Conditioned on the defect profile $\mathcal{D}(\Omega_p^{(k)})$, a redesigned architecture is constructed by modifying only the permissible implementation sites within the modular interface, thereby ensuring compatibility and isolating architectural changes to the designated design space. The design step produces $S$ variants by instantiating distinct

*Table 2.* Performance of various agent frameworks on the WebWalkerQA, xBench-DS, TaskCraft, and GAIA benchmarks.

| Framework | Model Family | WebWalker QA | xBench -DS | Task Craft | GAIA | | | |
|---|---|---|---|---|---|---|---|---|
| | | | | | Avg. | Level 1 | Level 2 | Level 3 |
| OWL Workforce (pass@3) | GPT-4o+o3-mini | 57.64 | 55.0 | 58.33 | 60.61 | 81.14 | 58.14 | 26.92 |
| OWL RP (pass@3) | GPT-4o+o3-mini | - | - | - | 58.18 | 81.14 | 54.65 | 23.08 |
| TapeAgents | Claude 3.7 *etc.* | - | - | - | 55.76 | 71.70 | 53.49 | 30.77 |
| AutoAgent | Claude 3.5 *etc.* | - | - | - | 55.15 | 71.70 | 53.40 | 26.92 |
| Smolagents 🤗 | GPT-4.1 | - | - | - | 55.15 | 67.92 | 53.49 | 34.62 |
| Smolagents 🤗 | GPT-5-mini | 58.82 | 51.0 | 64.00 | 55.75 | 69.81 | 54.65 | 30.77 |
| Magnetic-1 | OpenAI o1 *etc.* | - | - | - | 46.06 | 56.60 | 46.51 | 23.08 |
| Cognitive Kernel-Pro (pass@1) | Claude-3.7 *etc.* | 60.64 | 56.0 | 66.00 | 60.00 | 79.25 | 56.98 | 30.77 |
| Cognitive Kernel-Pro (pass@3) | Claude-3.7 *etc.* | - | - | - | 75.15 | 84.91 | 73.26 | 61.54 |
| OAgents | Claude-3.7 *etc.* | 58.23 | 47.0 | - | 66.67 | 77.36 | 66.28 | 46.15 |
| JoyAgents | Claude-4, o4-mini | - | - | - | 75.2 | 86.8 | 77.9 | 42.3 |
| Agent KB (pass@1) | GPT-4.1 | 60.59 | 48.0 | 61.67 | 61.21 | 79.25 | 58.14 | 34.62 |
| Agent KB (pass@2) | GPT-4.1 | 68.82 | 58.0 | 72.67 | 67.27 | 83.02 | 67.44 | 34.62 |
| Agent KB (pass@3) | GPT-4.1 | 73.53 | 68.0 | 75.33 | 73.94 | 84.91 | 73.26 | 53.85 |
| Flash-Searcher ⚡ (pass@1) | GPT-5-mini | 71.18 | 69.0 | 69.67 | 69.09 | 79.25 | 69.77 | 46.15 |
| Flash-Searcher ⚡ (pass@1) | Kimi K2 | 52.35 | 66.0 | 58.00 | 52.12 | 58.49 | 52.33 | 34.62 |
| Flash-Searcher ⚡ (pass@1) | DeepSeek V3.2 | 69.41 | 68.0 | 69.33 | 60.61 | 79.25 | 53.49 | 46.15 |
| `MemEvolve` + 🤗 (pass@1) | GPT-5-mini | 61.18 | 57.0 | 67.67 | 64.24 | 83.02 | 58.14 | 46.15 |
| `MemEvolve` + 🤗 (pass@2) | GPT-5-mini | 67.06 | 63.0 | 75.00 | 67.88 | 84.91 | 63.95 | 46.15 |
| `MemEvolve` + 🤗 (pass@3) | GPT-5-mini | 71.18 | 68.0 | 77.00 | 72.12 | 88.68 | 68.60 | 50.00 |
| `MemEvolve` + ⚡ (pass@1) | GPT-5-mini | 74.71 | 74.0 | 72.00 | 73.33 | 83.02 | 73.26 | 53.85 |
| `MemEvolve` + ⚡ (pass@2) | GPT-5-mini | 79.41 | 77.0 | 75.00 | 77.58 | 92.45 | 74.42 | 57.69 |
| `MemEvolve` + ⚡ (pass@3) | GPT-5-mini | 81.18 | 78.0 | 79.33 | 80.61 | 94.34 | 79.07 | 57.69 |
| `MemEvolve` + ⚡ (pass@1) | Kimi K2 | 69.41 | 68.0 | 68.00 | 61.21 | 67.92 | 63.95 | 38.46 |
| `MemEvolve` + ⚡ (pass@1) | DeepSeek V3.2 | 72.35 | 70.0 | 72.67 | 67.88 | 83.02 | 63.95 | 50.00 |

but valid configurations of the four components:

$$\Omega_{p,s}^{(k+1)} = \text{Design}\Big(\Omega_p^{(k)}, \mathcal{D}(\Omega_p^{(k)}), s\Big), \ s \in \{1, \dots, S\}. \quad (9)$$

These variants differ in encoding strategies, storage rules, retrieval constraints, or management policies, yet all conform to the unified memory-system interface and remain executable by the agent.

**Resulting update.** Aggregating all descendants across parents yields the next set of candidate architectures:

$$\{\Omega_{j'}^{(k+1)}\}_{j' \in \mathcal{J}^{(k+1)}} = \bigcup_{\Omega_p^{(k)} \in \mathcal{P}^{(k)}} \{\Omega_{p,s}^{(k+1)}\}_{s=1}^{S}. \quad (10)$$

This diagnose-and-design evolution operationalizes $\mathcal{F}$ for producing increasingly adaptive memory systems, ensuring that architectural updates are both empirically grounded and structurally constrained within the unified design space.

## 5. Experiments

### 5.1. Experiment Setup

**Benchmarks** We evaluate the proposed framework across four challenging agentic benchmarks, including *GAIA* (Mi-

alon et al., 2023), *WebWalkerQA* (Wu et al., 2025a), *xBench-DeepSearch* (xBench-DS) (Chen et al., 2025), as well as *TaskCraft* (Shi et al., 2025a). Further statistics and details are provided in Section B.1.

**Method Configurations** We run the dual-evolution process for $K_{\max} = 3$ iterations. In the outer loop, the survivor budget is set as $K = 1$; at each iteration, only the top-ranked architecture is retained and expanded to $S = 3$ descendants. In the inner loop, each candidate architecture $\Omega_j^{(k)}$ is evaluated on a batch $\mathcal{T}_j^{(k)}$ of 60 task trajectories, consisting of 40 newly sampled tasks and 20 tasks reused from the previous iteration to stabilize inter-iteration comparison.

**Agent Framework** We integrate `MemEvolve` into two representative agentic frameworks: 🤗 *SmolAgent* (Roucher et al., 2025), a lightweight two-agent architecture, and ⚡ *Flash-Searcher* (Qin et al., 2025), a high-performance single-agent deep research system. To assess the generalization and plug-and-play capability of `MemEvolve`, we further evaluate it on two held-out multi-agent systems: Tencent's 🌐 *Cognitive Kernel-Pro (CK-Pro)* (Fang et al., 2025c), a three-agent framework comprising main/file/web agents; and 🦉 *OWL* (Hu et al., 2025b), a hierarchical system including planner, coordinator, web, document, and coding

*Table 3.* Performance, cost, delay, and steps across datasets under different memory settings for ⚡ Flash-Searcher. Here, *cost* denotes the average API cost incurred per task query, *delay* measures the average execution latency (seconds) per task, and *#steps* reports the number of agent interaction steps required to complete each task.

| Memory Setting | GAIA | | | | xBench | | | | WebWalkerQA | | | |
|---|---|---|---|---|---|---|---|---|---|---|---|---|
| | Perf. | Cost | Delay | #Steps | Perf. | Cost | Delay | #Steps | Perf. | Cost | Delay | #Steps |
| No-Memory | 69.09 | 0.086 | 505.46 | 10.44 | 69.00 | 0.141 | 523.05 | 14.69 | 71.18 | 0.048 | 251.57 | 6.91 |
| Generative | 66.67 | 0.061 | 436.26 | 8.87 | 70.00 | 0.131 | 818.37 | 13.45 | 72.35 | 0.045 | 268.56 | 6.64 |
| Voyager | 69.70 | 0.060 | 499.89 | 9.25 | 68.00 | 0.117 | 553.46 | 12.71 | 73.53 | 0.049 | 333.69 | 6.99 |
| DILU | 66.67 | 0.059 | 444.62 | 8.91 | 69.00 | 0.134 | 500.72 | 13.83 | 72.94 | 0.046 | 272.16 | 6.96 |
| ExpeL | 66.06 | 0.059 | 500.11 | 8.68 | 64.00 | 0.123 | 710.32 | 13.05 | 69.41 | 0.076 | 385.28 | 10.96 |
| AWM | 67.27 | 0.062 | 584.88 | 10.23 | 71.00 | 0.138 | 761.33 | 14.12 | 72.35 | 0.068 | 397.20 | 11.40 |
| Mobile-E | 69.09 | 0.065 | 321.80 | 9.35 | 68.00 | 0.120 | 537.18 | 13.16 | 71.76 | 0.059 | 296.01 | 6.52 |
| Cheatsheet | 68.48 | 0.069 | 559.81 | 9.72 | 65.00 | 0.174 | 818.07 | 15.99 | 72.94 | 0.057 | 367.13 | 7.59 |
| MemEvolve | 73.33 | 0.085 | 693.33 | 10.14 | 74.00 | 0.136 | 773.06 | 14.20 | 74.71 | 0.040 | 332.49 | 6.64 |

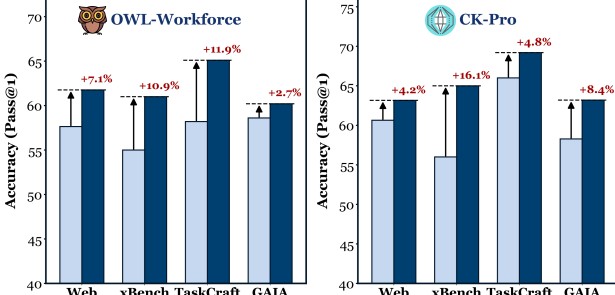

*Figure 3.* The cross-framework generalization analysis. We transfer the memory system evolved on TaskCraft+⚡ to 🦉 and 🌐. Red percentages denote the relative score gains of each framework after integrating MemEvolve over its memory-free counterpart.

agents. This diversity in architecture and system complexity enables a comprehensive examination of the adaptability of MemEvolve across heterogeneous agentic scaffolds.

**Model Configurations** We instantiate MemEvolve using GPT-5-MINI (OpenAI, 2025) as the LLM backbone for the underlying agentic frameworks, and for supporting the meta-evolution operator $\mathcal{F}(\cdot)$. To further evaluate the cross-LLM generalization capability of MemEvolve, we additionally consider alternative backbones, including DEEPSEEK V3.2 (DeepSeek-AI et al., 2025), and KIMI K2 (Team et al., 2025a). We explicitly report the specific LLM backbone used by each framework in the following experiments.

**5.2. Main Results**

We report the pass@1–3 performance of MemEvolve integrated with SmolAgent and Flash-Searcher in Table 2, together with its generalization results when paired with unseen LLMs (KIMI K2, DEEPSEEK V3.2). Notably, on the relatively simple TaskCraft, we evolve two distinct memory systems using MemEvolve+🦊 and MemEvolve+⚡, respectively. These evolved memory systems are then fixed and evaluated on WebWalkerQA and xBench-DS, *i.e.*, without conducting dataset-specific meta-evolution.

MemEvolve **Exhibits Cross-Task, Cross-Model, and Cross-Framework Generalization.** As shown in Table 2, memory architecture is a primary determinant of agent performance and transferability. On xBench, 🦊+GPT-5-MINI attains a pass@1 of 51%, which increases by 6% and reaches 68.0% at pass@3 after integrating MemEvolve; similarly, ⚡+GPT-5-MINI improves from 69% to 74%. In contrast to prior approaches that rely on task-specific tuning or static memory pipelines, the memories evolved by MemEvolve transfer without dataset-level meta-evolution: memory systems learned on TaskCraft yield consistent gains on harder benchmarks (WebWalkerQA+🦊: 58.82 → 61.18%; xBench+⚡: 69.0 → 74.0%), indicating task-agnostic design principles rather than dataset overfitting. Beyond cross-task transfer, MemEvolve exhibits strong cross-LLM generalization: although meta-evolution is conducted with GPT-5-MINI, the evolved memories transfer effectively to KIMI K2 and DEEPSEEK V3.2 without manual adaptation, with KIMI K2+⚡ achieving gains of 17.06% on WebWalkerQA and 10.0% on TaskCraft. Finally, as shown in Figure 3, directly plugging these memories into heterogeneous agentic frameworks (*e.g.*, 🦉, 🌐) consistently improves performance despite substantial architectural differences, underscoring that MemEvolve learns framework-agnostic, reusable memory abstractions rather than narrowly task-bound heuristics.

**5.3. Self-Evolving Memory Comparison**

We further compare the memory systems automatically evolved by MemEvolve against prevailing human-designed self-improving memory systems. In Table 3, we integrate seven representative self-improving memory systems implemented in EvolveLab with Flash-Searcher, and comprehensively report performance, per-task cost/execution latency/execution steps. Results for MemEvolve are obtained using the system evolved on TaskCraft+⚡+GPT-5-MINI.

MemEvolve **Delivers Robust and Consistent Improvements.** Despite faithful re-implementations, many exist-

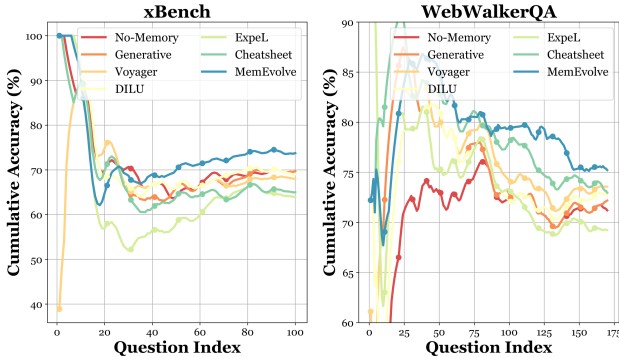

*Figure 4.* Evolution of cumulative accuracy across question indices. Cumulative accuracy at index $i$ is defined as the average accuracy over the first $i$ questions. The curves exhibit larger fluctuations at early indices due to limited sample size, and gradually stabilize as more questions are accumulated.

ing memory systems fail to deliver consistent improvements across benchmarks. For instance, DILU improves xBench and WebWalkerQA yet degrades GAIA by $2.42\%$, while Dynamic Cheatsheet yields a modest $1.76\%$ gain on WebWalkerQA but performs poorly on GAIA and xBench. ExpeL underperforms across all three, reflecting its original design for relatively simple embodied or QA settings (*e.g.*, ALFWorld, HotpotQA) and its limited suitability for long-horizon, long-context deep research. In contrast, `MemEvolve` produces stable, benchmark-agnostic gains: although evolved on TaskCraft, it consistently improves performance by $3.54\% \sim 5.0\%$ across all evaluated benchmarks. These improvements are achieved without substantial cost inflation, maintaining API expenses comparable to the No-Memory baseline (*e.g.*, GAIA: $0.085 vs. $0.086; xBench: $0.136 vs. $0.141) and execution delays on par with other self-improving methods (*e.g.*, GAIA: $693.33s$ vs. $584.88s$ for AWM and $559.81s$ for Cheatsheet; xBench: $773.06s$ vs. $761.33s$ for AWM and $818.07s$ for Cheatsheet). As shown in Figure 4, while early-stage performance exhibits higher variance, `MemEvolve` converges to a consistently superior regime, indicating principled, transferable memory designs rather than brittle, task-specific heuristics. Three-run evaluations in Table 5 yield consistent rankings under repeated trials.

## 5.4. Framework Analysis

**Ablation Study** We isolate the contribution of each memory module by evolving only one component at a time while freezing the other three components at the R0 AgentKB initialization. As shown in Table 4, evolving retrieval alone yields the largest single-module gain ($+10.00$), consistent with the qualitative observation that early evolution shifts retrieval from static semantic matching toward agent-gated, context-aware filtering. Nevertheless, no single component matches the full framework: joint evolution improves

*Table 4.* Component-level ablation on TaskCraft. Each "Only" variant evolves one memory module while keeping the remaining modules fixed at the R0 AgentKB initialization; Full `MemEvolve` jointly evolves all modules.

| Round | Only Enc. | Only Ret. | Only Mng. | Only Store | Full |
|---|---|---|---|---|---|
| R0 | 61.67 | 61.67 | 61.67 | 61.67 | 61.67 |
| R1 best | 63.33 | 66.67 | 63.33 | 63.33 | 70.00 |
| R2 best | 66.67 | 71.67 | 65.00 | 63.33 | 75.00 |
| R3 best | 68.33 | 71.67 | 66.67 | 65.00 | **78.33** |
| $\Delta$ | +6.66 | +10.00 | +5.00 | +3.33 | **+16.66** |

TaskCraft by $+16.66$, indicating that the final gains arise from cross-module coupling, such as improved encoders producing memory entries that are more effectively exploited by adaptive retrieval and management.

**Sensitivity Analysis** The default evolutionary budget ($K = 1, S = 3$) follows a compact $(1 + \lambda)$-style strategy: one survivor seeds the next generation, while three descendants provide limited but structured exploration under expensive agent-level fitness evaluation. We further examine sensitivity to the evolution horizon by extending the process to five rounds in Table 6. Performance continues to improve from R3 to R5 on both in-domain TaskCraft ($78.33 \rightarrow 80.00$) and OOD WebWalkerQA ($76.27 \rightarrow 77.65$), while cost and latency decrease further. This supports $K_{\max} = 3$ as an efficient default while suggesting that longer evolution can further optimize resource efficiency. Cross-backbone and search-baseline analyses are provided in Tables 7 and 8.

## 5.5. Meta-Evolving Dynamics

Having established the performance gains of `MemEvolve`, we examine how meta-evolution proceeds in practice and which components are modified. As shown in Figure 6, `MemEvolve` initializes from the predefined AgentKB structure and iteratively evolves toward more efficient memory architectures. Figures 8 and 9 highlight two high-performing systems discovered along this trajectory, RIVA and CEREBRA, while Figure 7 presents a system evolved from a minimal few-shot baseline, denoted LIGHTWEIGHT.

**Agents Spontaneously Evolve Efficient Memory Architectures** Starting from AgentKB with frozen encoding and storage, `MemEvolve` explores a range of evolutionary directions, from aggressive designs (*e.g.*, $\Omega_1^{(1)}$, which decomposes trajectories into 9 fine-grained skills) to conservative ones (*e.g.*, $\Omega_3^{(1)}$, which stores trajectories at four levels and applies an LLM guardrail during retrieval), with the latter winning in the first round. This stage is marked by increasing *agentic* control, as encoding and retrieval shift from predefined pipelines to agent-driven decisions. By the third round, the transition from RIVA to CEREBRA introduces two advances: distillation of reusable tools/insights and periodic database maintenance, yielding faster and more stable evolutionary progress for downstream agents.

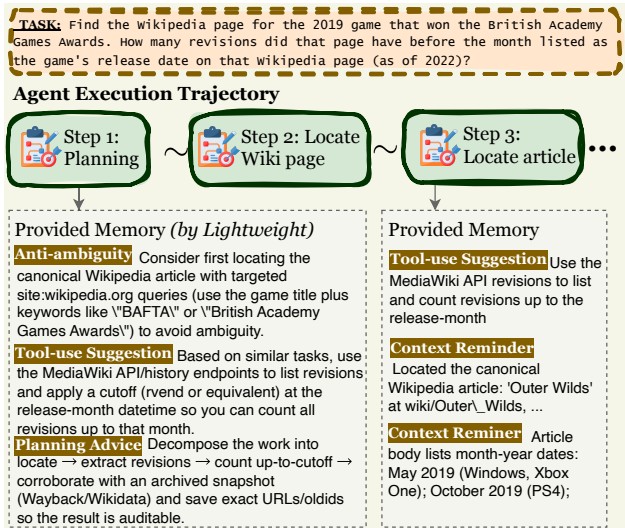

*Figure 5.* Illustration of how evolved memories are instantiated. The memory system adaptively provides stage-specific guidance, ranging from high-level planning and task decomposition to fine-grained tool-use suggestions and salient context recall.

**Evolved Memory Systems Are Effective in Practice** Qualitative examples from LIGHTWEIGHT (Figure 5) demonstrate adaptive memory behavior across task phases. During planning, the system provides high-level guidance for task decomposition; during execution, it supplies fine-grained tool-use recommendations and maintains a working memory that highlights salient prior information. These illustrate the practical utility of the evolved memory systems.

## 6. Conclusion

This work introduces a unified design space and standardized codebase, `EvolveLab`, for self-evolving agent memory, upon which we build `MemEvolve`, a meta-evolutionary memory framework. Departing from manually crafting a single memory architecture and assuming cross-domain generalization, `MemEvolve` enables adaptive, memory architecture evolution in bi-level optimization. Extensive evaluations across diverse benchmarks and LLMs demonstrate its effectiveness, robustness, and generalization.

## Impact Statement

**Ethical Considerations.** This work studies meta-evolving memory architectures for LLM-based agents and evaluates the proposed framework exclusively on publicly available benchmarks and synthetic or controlled interaction trajectories. The method does not involve the collection of personal data, nor does it require access to sensitive, private, or proprietary information. We therefore do not identify specific ethical risks beyond the well-known considerations associated with deploying large language models, such as potential misuse or overreliance on automated systems.

**Societal Impact.** By enabling memory systems that adapt to task demands and transfer across models and frameworks, this work aims to improve the robustness, efficiency, and generalization of agentic systems in long-horizon and complex settings. More reliable and transparent agent behavior may benefit applications in research assistance and software development. We expect the primary societal impact to be positive, facilitating more responsible design, evaluation, and deployment of self-improving AI systems.

## Acknowledgements

This work was supported by the National Natural Science Foundation of China under Grant No. 62320106007, and by NUS Grant A-0010106-00-00, A-8004365-00-00 and A-8004410-01-00.

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

# A. EvolveLab Implementation

**EvolveLab** is designed as a modular and extensible codebase to support the systematic study of self-evolving agent memory systems. It provides a unified interface that abstracts the complexities of diverse memory architectures, enabling standardized implementation, evaluation, and meta-evolution.

## A.1. Unified Interface and Abstract Base Class

The cornerstone of **EvolveLab** is the BaseMemoryProvider abstract base class (ABC), which defines the fundamental protocol for all memory systems. As shown in the code snippet below, the interface enforces two primary operations that map to the modular design space (*Encode, Store, Retrieve, Manage*):

- **Retrieve (provide_memory)**: Handles context-aware memory recall. It accepts a MemoryRequest containing the current task query, execution context, and system status, and returns a MemoryResponse containing a list of relevant MemoryItems.
- **Encode & Store (take_in_memory)**: Orchestrates the ingestion of new experiences. This method processes a TrajectoryData object, which encapsulates the complete history of a task execution, extracts structural insights or tools (*Encode*), and persists them into the underlying storage medium (*Store*).

While take_in_memory primarily integrates the *Encode* and *Store* stages, the *Manage* functionality that is responsible for offline consolidation or selective forgetting is typically implemented as auxiliary methods within the provider classes or invoked during specific lifecycle events.

```python
class BaseMemoryProvider(ABC):
    """Abstract base class for memory providers"""

    def __init__(self, memory_type: MemoryType, config: Optional[dict] = None):
        self.memory_type = memory_type
        self.config = config or {}

    @abstractmethod
    def provide_memory(self, request: MemoryRequest) -> MemoryResponse:
        """
        Retrieve relevant memories based on query, context and status
        Args:
            request: MemoryRequest containing query, context, status and optional params
        Returns:
            MemoryResponse containing relevant memories
        """
        pass

    @abstractmethod
    def take_in_memory(self, trajectory_data: TrajectoryData) -> tuple[bool, str]:
        """
        Store/ingest new memory from trajectory data
        Args:
            trajectory_data: TrajectoryData containing query, trajectory and metadata
        Returns:
            tuple[bool, str]: (Success status of memory ingestion, Description of absorbed
     memory)
        """
        pass

    @abstractmethod
    def initialize(self) -> bool:
        """
        Initialize the memory provider (load existing data, setup indices, etc.)
        Returns:
            bool: Success status of initialization
        """
        pass
```

```
38
39     def get_memory_type(self) -> MemoryType:
40         """Get the type of this memory provider"""
41         return self.memory_type
42
43     def get_config(self) -> dict:
44         """Get the configuration of this memory provider"""
45         return self.config.copy()
```

*Listing 1.* The Abstract Base Class of Memory Providers

## A.2. Standardized Data Carriers

To ensure seamless interoperability across heterogeneous memory designs and agent frameworks, `EvolveLab` utilizes standardized memory data carriers. These structures act as the "universal language" of the framework:

- **`MemoryItem`**: The fundamental unit of information, capable of representing raw text, distilled insights, or executable code (APIs). Each item includes metadata such as creation timestamps, confidence scores, and source identifiers.
- **`TrajectoryData`**: A comprehensive container for task execution history, including the initial query, full interaction traces (state-action pairs), and terminal rewards. It serves as the raw substrate for memory evolution.
- **`MemoryRequest/Response`**: Standardized envelopes for retrieval queries and results, ensuring that any agent system can interact with any memory provider without architecture-specific modifications.

## A.3. Implementation Examples: ExpeL and SkillWeaver

The versatility of the `EvolveLab` interface is demonstrated by our implementation of twelve distinct memory systems. Two representative examples are:

- **ExpeLProvider**: Implements a contrastive learning-based memory. Its `take_in_memory` function identifies successful and failed trajectories to distill high-level "insights" into a textual format. These insights are stored in a vector database and retrieved via semantic similarity during `provide_memory` to guide the agent away from previous mistakes.
- **SkillWeaverProvider**: Operates in a tool-centric design space. Its `take_in_memory` logic uses an LLM to synthesize reusable Python functions (skills) from successful trajectories. These skills are stored as executable code-level repositories and are dynamically retrieved and injected into the agent's action space through the unified `MemoryItem` interface.

## A.4. Evaluation

Beyond unified implementation, `EvolveLab` provides a standardized testbed for rigorously assessing memory architectures across diverse agentic tasks. The framework offers out-of-the-box support for multiple challenging benchmarks, including GAIA (Mialon et al., 2023), xBench (Chen et al., 2025), and WebWalkerQA (Wu et al., 2025a). `EvolveLab` accommodates two evaluation paradigms: an ■ **online** mode, where the experiential memory base is updated on-the-fly as the agent system processes a continuous stream of tasks, and an ■ **offline** mode, where the memory system first accumulates experience from a static set of trajectories before being assessed on separate, unseen tasks. To ensure robust and versatile assessment, we support multiple evaluation protocols, including exact string matching and flexible LLM-as-a-Judge.

# B. Experiment Details

## B.1. Dataset Details

The four datasets used in this study are described and summarized as follows:

- *GAIA* (Mialon et al., 2023) consists of 165 tasks, including 53 Level-1, 86 Level-2, and 26 Level-3 problems. For evaluating `MemEvolve` on GAIA+😭 and GAIA+⚡, the memory systems are evolved using GAIA Level-1 tasks together with 67 TaskCraft queries. Meta-evolution is conducted for three rounds, with 40 trajectories per round.

*Table 5.* Three-run robustness evaluation for `MemEvolve` and competitive self-improving memory baselines in Table 3. We report mean accuracy and standard deviation.

| Method | GAIA | WebWalkerQA | xBench |
|---|---|---|---|
| Voyager | $69.90 \pm 0.76$ | $73.14 \pm 1.00$ | $68.00 \pm 1.63$ |
| AWM | $67.68 \pm 1.51$ | $72.16 \pm 0.28$ | $70.33 \pm 0.47$ |
| Mobile-E | $68.69 \pm 1.03$ | $71.76 \pm 1.44$ | $68.33 \pm 1.25$ |
| `MemEvolve` | $\mathbf{73.94} \pm 0.86$ | $\mathbf{76.27} \pm 1.47$ | $\mathbf{73.67} \pm 0.47$ |

*Table 6.* Extended evolution beyond the default three rounds. The OOD WebWalkerQA column evaluates the best architecture from each round without re-evolution.

| Round | TaskCraft | Cost | Delay | OOD WebWalkerQA |
|---|---|---|---|---|
| R0 | 61.67 | 0.092 | 485.25 | $65.10 \pm 2.65$ |
| R1 best | 70.00 | 0.105 | 512.40 | $67.06 \pm 0.96$ |
| R2 best | 75.00 | 0.073 | 355.40 | $70.20 \pm 1.47$ |
| R3 best | 78.33 | 0.056 | 364.37 | $76.27 \pm 1.47$ |
| R4 best | 80.00 | 0.048 | 318.52 | $76.86 \pm 1.94$ |
| R5 best | 80.00 | 0.043 | 295.60 | $77.65 \pm 1.44$ |

*Table 7.* Cross-backbone transfer with Claude-3.7-Sonnet. The `MemEvolve` memory architecture is evolved on TaskCraft with GPT-5-Mini and transferred to Claude-3.7-Sonnet without re-evolution.

| Framework | Model | WebWalkerQA | xBench-DS | TaskCraft | GAIA Avg. |
|---|---|---|---|---|---|
| OAgents | Claude-3.7-Sonnet | 58.23 | 47.00 | – | 66.67 |
| CK-Pro (pass@1) | Claude-3.7-Sonnet | 60.64 | 56.00 | 66.00 | 60.00 |
| CK-Pro (pass@3) | Claude-3.7-Sonnet | 65.88 | 67.00 | 70.66 | 75.15 |
| `MemEvolve` +Flash (pass@1) | Claude-3.7-Sonnet | 71.76 | 69.00 | 69.33 | 69.69 |
| `MemEvolve` +Flash (pass@3) | Claude-3.7-Sonnet | **77.05** | **74.00** | **75.33** | **78.78** |

- *WebWalkerQA* (Wu et al., 2025a) evaluates an agent's ability to handle complex, multi-turn web interactions, comprising 680 real-world queries across four domains and over 1,373 webpages. We sample a subset of 170 queries for evaluation, with the sampling script released in our codebase. All memory systems used for WebWalkerQA are meta-evolved on TaskCraft.

- *xBench-DeepSearch* (xBench-DS) (Chen et al., 2025) contains 100 tasks that assess agentic planning, tool use, and reasoning. Similar to WebWalkerQA, the memory systems used for xBench-DS evaluation are entirely meta-evolved on TaskCraft.

- *TaskCraft* (Shi et al., 2025a) is a synthetic benchmark generated via an autonomous data pipeline. We collect 300 queries as a working subset and use 120 of them for three rounds of meta-evolution, with 40 queries per round. Meta-evolution for 🫠 and ⚡ is performed independently.

## B.2. Robustness Analyses

We report robustness analyses on repeated trials, longer evolution horizons, cross-backbone transfer, and search baselines.

**Statistical Significance Testing** We repeat representative evaluations three times for `MemEvolve` and competitive memory baselines. As shown in Table 5, `MemEvolve` maintains the highest mean performance across all three benchmarks with low cross-run variance.

**Extended Evolution** We extend the evolution horizon from three to five rounds to test sensitivity to $K_{\max}$. Table 6 shows that later rounds preserve the accuracy gains while further reducing cost and delay.

**Cross-Backbone Transfer** We evaluate transfer to Claude-3.7-Sonnet under the same fixed evolved memory architecture. As reported in Table 7, the transferred memory remains effective under this unseen backbone.

**Search Baselines** We compare diagnose-and-design evolution with random architecture composition and greedy component substitution. Table 8 indicates that structured diagnosis yields larger and more stable gains than both search baselines.

*Table 8.* Search-baseline comparison. All methods start from the same AgentKB initialization. Greedy Search enumerates eight single-module substitutions per round and accepts the locally best variant; `MemEvolve` uses trajectory-grounded diagnosis to generate three holistic descendants. WebWalkerQA is evaluated OOD without re-evolution.

| Round | MemEvolve | | Random Search | | Greedy Search | |
|---|---|---|---|---|---|---|
| | **TaskCraft** | **WebWalkerQA** | **TaskCraft** | **WebWalkerQA** | **TaskCraft** | **WebWalkerQA** |
| R0 | 61.67 | 65.10 | 61.67 | 65.10 | 61.67 | 65.10 |
| R1 best | 70.00 | 67.06 | 63.33 | 64.51 | 65.00 | 65.88 |
| R2 best | 75.00 | 70.20 | 66.67 | 67.84 | 68.33 | 68.24 |
| R3 best | 78.33 | 76.27 | 65.00 | 65.88 | 70.00 | 70.59 |
| Δ | **+16.66** | **+11.17** | **+3.33** | **+0.78** | **+8.33** | **+5.49** |

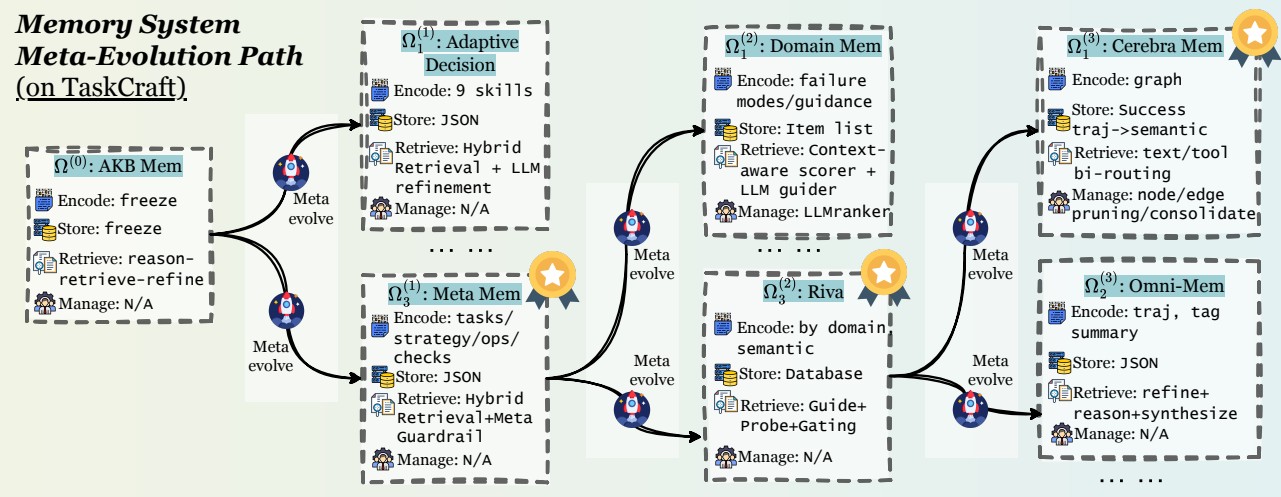

*Figure 6.* Illustration of the progressive evolution from the fixed AgentKB architecture to increasingly agentic and efficient memory architectures. Each stage reflects structural and functional modifications in memory encoding, storing, retrieval, and maintenance, culminating in high-performing systems such as Riva and Cerebra.

### B.3. Evolution Dynamics

We illustrate the progressive evolution from the fixed AgentKB architecture to increasingly agentic and efficient memory architectures of `MemEvolve` in Figure 6.

## C. Memory System Demonstration

To provide a concrete and intuitive understanding of the memory architectures evolved by `MemEvolve`, we visualize three representative systems discovered along different evolutionary trajectories, as shown in Figures 7 to 9. These examples highlight how `MemEvolve` progressively transforms simple, static memory mechanisms into more expressive and adaptive architectures by modifying memory encoding, retrieval, and management strategies. Together, they illustrate the diversity of memory designs that can emerge under the same meta-evolutionary framework.

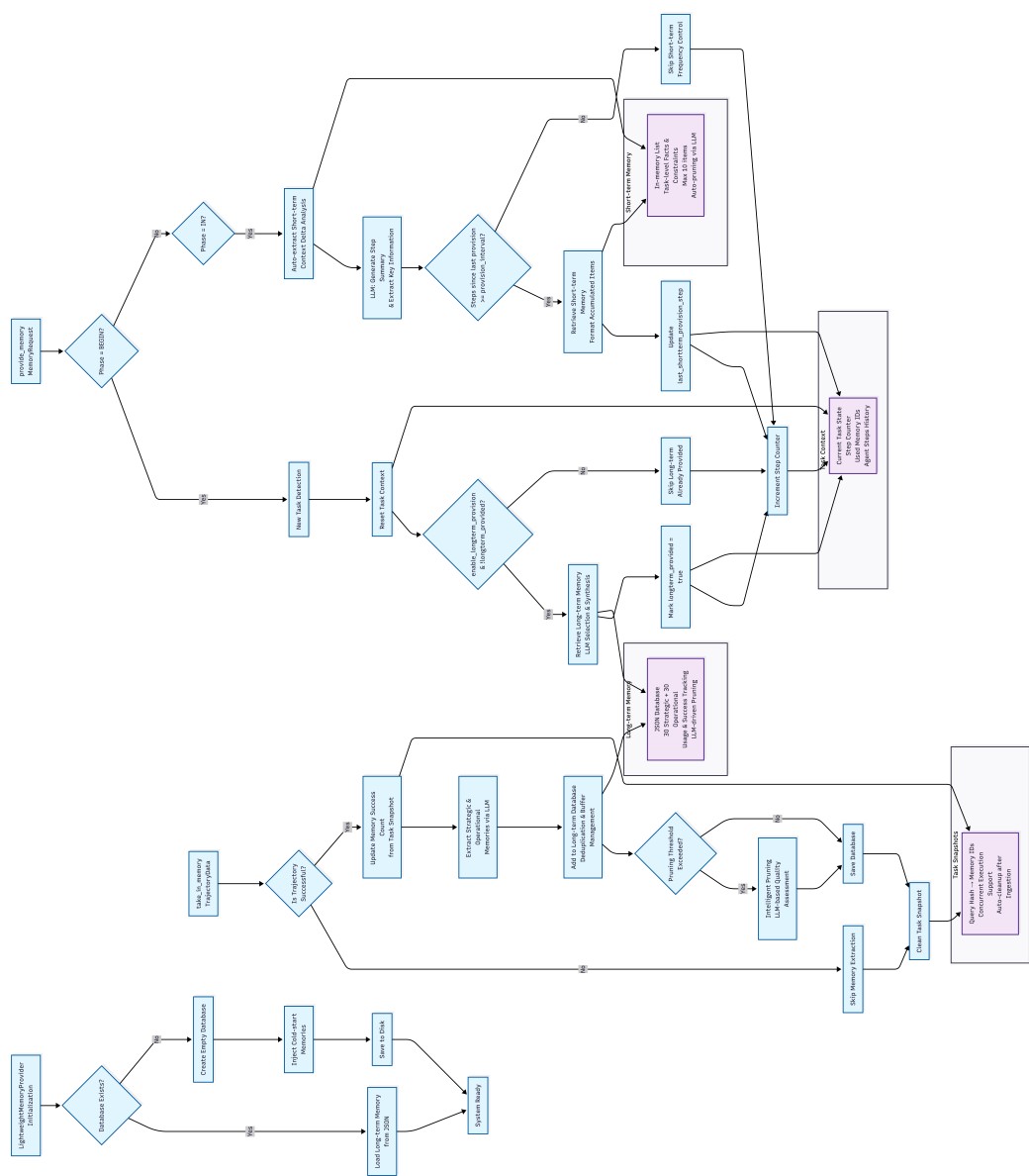

*Figure 7.* Illustration of the LIGHTWEIGHT memory system evolved by `MemEvolve`. The evolutionary starting point is a minimal few-shot trajectory memory, similar to MEMORYBANK, where each completed trajectory is stored verbatim. For a new task, the agent retrieves the top-$k$ most similar trajectories via vector similarity and directly conditions on them. `MemEvolve` progressively refines this baseline into a more structured and stage-aware memory system.

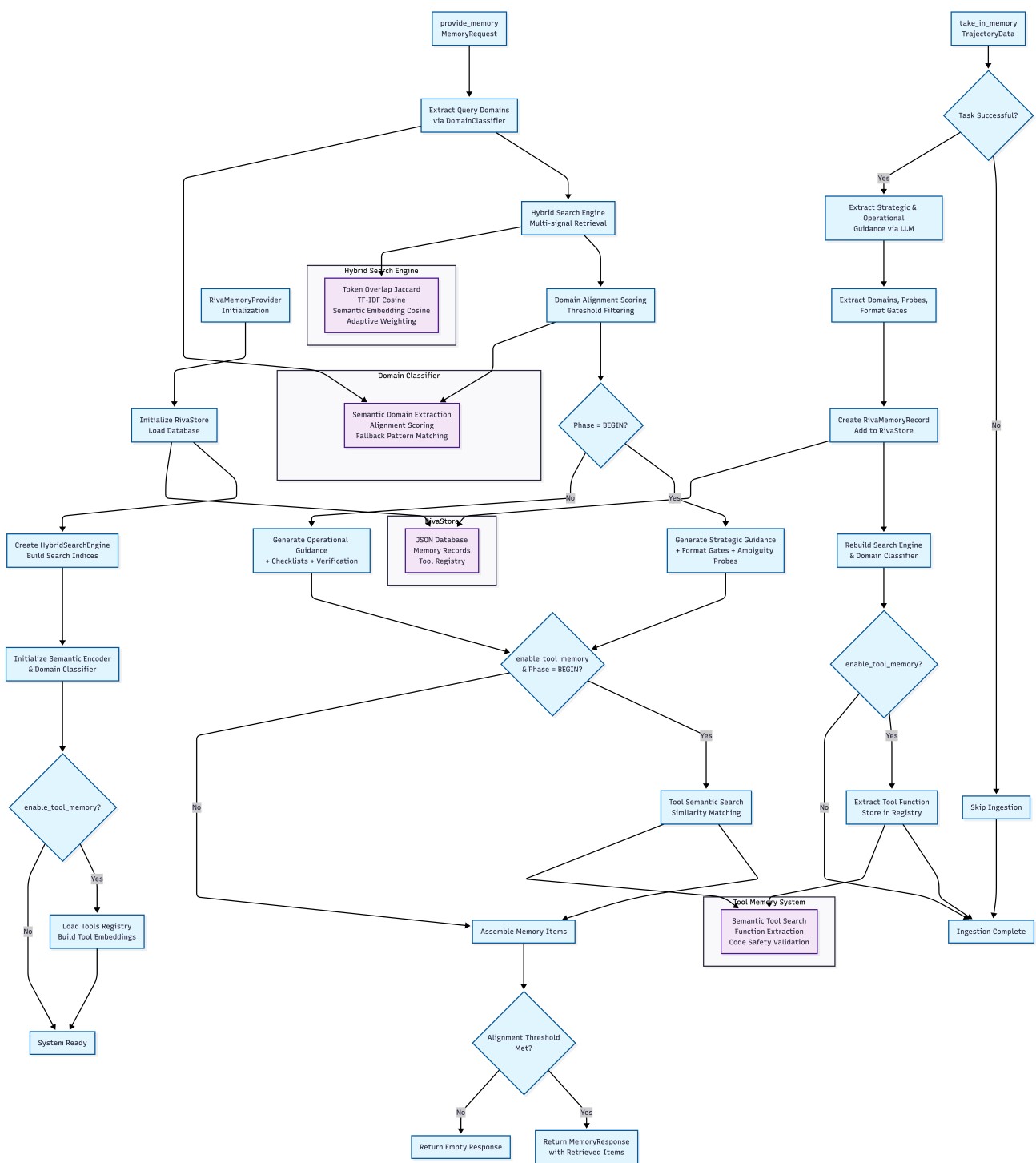

*Figure 8.* Illustration of the RIVA memory system evolved by `MemEvolve`. Its evolutionary initialization follows an AGENTKB-style architecture, but without inheriting the large and costly offline knowledge base. Through meta-evolution, RIVA develops more agent-centric encoding and retrieval strategies while remaining lightweight and fully online.

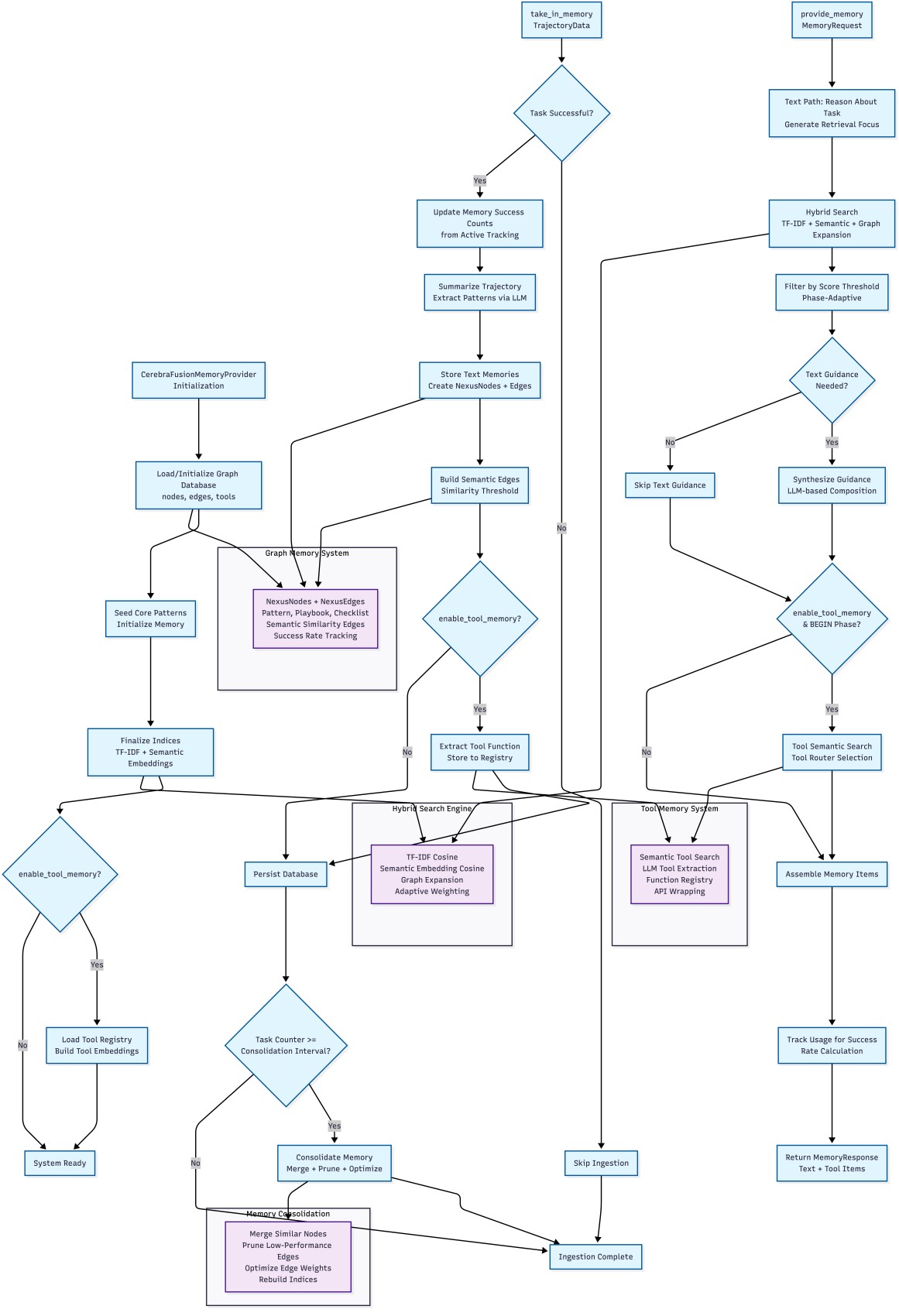

*Figure 9.* Illustration of the CEREBRA memory system evolved by `MemEvolve`. Starting from the same AGENTKB-style initialization (without the offline knowledge base), CEREBRA further evolves to distill both reusable tools and abstract knowledge from experience, and incorporates working memory maintenance mechanisms to support long-horizon agent evolution.

