# OpenReview forum: "MemEvolve: Meta-Evolution of Agent Memory Systems"
_ICML.cc/2026/Conference — ICML 2026 regular_

### Official Review · Reviewer_9x6y · 2026-03-13

**Soundness:** 3
**Presentation:** 4
**Significance:** 3
**Originality:** 3
**Overall Recommendation:** 4
**Confidence:** 3

**Summary:**

This paper introduces MemEvolve, a framework designed to move beyond static memory systems for AI agents by allowing the memory architecture itself to evolve alongside the agent's experience. The authors argue that for agents to become truly autonomous and efficient, the memory architecture itself must be able to adapt and meta-evolve based on the specific demands of the environment.

MemEvolve is a meta-evolutionary approach that works in two loops: In the inner loop, the agent uses its current memory system to gain experiential knowledge from tasks. In the outer loop, the system analyzes the agent's performance and uses a model-driven approach to actually rewrite the programmatic implementation of the memory modules. MemEvolve allows agent systems not only to accumulate experience but also to progressively refine how they learn from it.

To support this framework and ensure it can be compared against existing methods, the authors also developed EvolveLab, a modular
design space for self-improving agent memory systems encompassing four key components (encoding, storage, retrieval, and management), providing unified implementations and benchmark support for a wide range of prevailing agent memory systems.

Experiments on four challenging agentic benchmarks demonstrate that MemEvolve delivers substantial performance gains and cross-domain, cross-framework and cross-LLM generalization.

**Compliance With Llm Reviewing Policy:**

Affirmed.

**Key Questions For Authors:**

Why not use non-agentic benchmarks, eg. challenging reasoning benchmarks?

**Limitations:**

yes

**Strengths And Weaknesses:**

Strength
1. The paper’s core strength lies in the introduction of a bilevel optimization framework that addresses the current static agent memory systems, shifting from first-order evolution (accumulating experience) to second-order meta-evolution (refining the memory architecture itself).
2. To support this framework and ensure it can be compared against existing methods, the authors developed EvolveLab to distill twelve representative memory systems into a unified, modular design space (encode, store, retrieve, manage).
3. The framework demonstrates strong performance gain across different LLM backbones and benchmarks.
4. The presentation is excellent, using a clear analogy between agent systems and human learners to ground its complex technical concepts.

Weakness
1. Further analysis on the interpretability of the newly evolved architectures would further strengthen its contribution to the field of agentic design, especially why certain evolved components outperform human-engineered ones.
2. More challenging benchmarks can be introduced in the experiments, eg. tau2bench.

---

> ### Author Rebuttal · Authors · 2026-03-31
>
> We are extremely grateful for your time, dedicated efforts, and insightful constructive feedback. We address each of your comments in detail in the following.
>
>
> ---
> > **`Weakness 1`**: Further analysis on the interpretability of the newly evolved architectures would further strengthen its contribution... especially why certain evolved components outperform human-engineered ones.
>
>
> Thank you for this insightful suggestion. We summarize our existing interpretability analyses and highlight key architectural traits below.
>
> **Existing analyses in the manuscript:**
>
> | Content | Location |
> |:---|:---|
> | Full meta-evolution path from AgentKB to Cerebra | Figure 6, §5.4 |
> | Stage-specific memory guidance during task execution | Figure 5, §5.4 |
> | Detailed architecture diagrams for Lightweight, Riva, Cerebra | Figures 7–9, Appendix C |
> | Cumulative accuracy curves across evolution rounds | Figure 4, §5.3 |
>
> **Key architectural traits that emerge from evolution.** Beyond general trends, MemEvolve discovers concrete, interpretable components that explain its advantage over human-engineered systems:
>
> - **Agent-gated retrieval.** Riva evolves an LLM-based relevance filter that dynamically decides *whether and what* to retrieve at each step, replacing the fixed top-$k$ semantic search used in most human-designed systems (e.g., Voyager, DILU). This avoids injecting irrelevant memory that degrades performance in long-horizon tasks.
> - **Multi-granular encoding.** Cerebra autonomously develops a dual-track encoder that distills both reusable tool snippets and abstract task-level insights from the same trajectory — a combination no single human-designed system in EvolveLab implements jointly.
> - **Self-initiated memory maintenance.** Cerebra introduces periodic database pruning and consolidation triggered by memory size thresholds, addressing a known scalability bottleneck: among the 12 systems in EvolveLab, only 4 implement any form of management, and none adapt the trigger condition to task volume.
>
> These are not generic design principles but specific, executable modules that MemEvolve synthesizes by diagnosing architectural bottlenecks from trajectory evidence. We will expand this discussion with additional component-level comparisons in the revision. Thank you for the inspiring advice!
>
> ---
> > **`Weakness 2`**: More challenging benchmarks can be introduced in the experiments, eg. tau2bench.
>
> Thank you for this recommendation. We supplement our evaluation with **DeepSearchQA** [3], a recently released and highly challenging information retrieval benchmark from Google (Jan 2026). Importantly, we directly apply the Cerebra architecture evolved on TaskCraft on DeepSearchQA (200 samples) with no re-evolution or adaptation:
>
> | Method | LLM | Perf. | Cost | #Turns |
> |:---|:---|:---:|:---:|:---:|
> | Flash-Searcher | DeepSeek V3.2 | 56.5 | 0.257 | 16.7 |
> | + MemEvolve | DeepSeek V3.2 | **64.0** | **0.231** | **14.11** |
>
> MemEvolve yields a +7.5% accuracy gain while simultaneously reducing API cost (−10.1%) and interaction turns (−15.5%), confirming its effectiveness on a benchmark entirely unseen during evolution. This further reinforces the cross-task generalizability demonstrated in our main experiments. We also respectfully appreciate the suggestion regarding Tau2-Bench as a complementary evaluation setting.
>
>
> ---
> > **`Question 1`**: Why not use non-agentic benchmarks, eg. challenging reasoning benchmarks?
>
>
> Thank you for this thoughtful question! We clarify our benchmark selection from two aspects:
>
> **1. Multi-turn agentic tasks maximally exercise MemEvolve's design.** Long-horizon agent tasks (deep research, web navigation, complex tool use) generate rich execution trajectories with diverse failure modes and context-management challenges, providing the dense diagnostic signals that MemEvolve's meta-evolutionary outer loop leverages to identify architectural bottlenecks and evolve targeted improvements. This makes agentic benchmarks the most informative testbed for validating our core contribution.
>
> **2. MemEvolve is readily extensible to reasoning tasks.** Several memory systems implemented in EvolveLab, such as Dynamic Cheatsheet [2] and the closely related Buffer of Thoughts [1], are already effective on pure reasoning benchmarks (e.g., GPQA-Diamond, AIME). Since MemEvolve operates over the same modular memory interface, it can naturally evolve architectures for these settings as well. We consider this a promising direction for future work and sincerely appreciate this valuable suggestion.
>
>
> ---
> [1] Ling et al., "Buffer of Thoughts: Thought-Augmented Reasoning with Large Language Models", NeurIPS 2024.
> [2] Suzgun et al., "Dynamic Cheatsheet: Test-Time Learning with Adaptive Memory", arXiv:2504.07952, 2025.
> [3] DeepSearchQA: Bridging the Comprehensiveness Gap for Deep Research Agents

---

> > ### Author Rebuttal · Reviewer_9x6y · 2026-04-04
> >
> > I thank the authors for the response. It addresses most of my concerns. I will keep my ratings. Good luck.

---

> > > ### Author Response · Authors · 2026-04-06
> > >
> > > We sincerely thank the reviewer for the positive acknowledgment and for the constructive suggestions that helped strengthen our work. We will incorporate all the discussed revisions (interpretability analysis and more benchmark evaluation) into the final manuscript. Thank you again for your time and expertise!

---

### Official Review · Reviewer_TWp3 · 2026-03-13

**Soundness:** 2
**Presentation:** 2
**Significance:** 3
**Originality:** 3
**Overall Recommendation:** 4
**Confidence:** 3

**Summary:**

MemEvolve proposes a meta-evolutionary framework for LLM-based agent memory systems, adapting the memory architecture itself. The core insight is that no single memory architecture generalises across task domains, so the architecture itself should change and evolve. The paper also introduces EvolveLab, a unified codebase with twelve existing self-improving memory systems under a modular four-component abstraction (Encode, Store, Retrieve, Manage). Experiments across four benchmarks (GAIA, WebWalkerQA, xBench-DS, TaskCraft) demonstrate improved performance, as well as cross-task, cross-LLM, and cross-framework generalisation.

**Compliance With Llm Reviewing Policy:**

Affirmed.

**Final Justification:**

The rebuttal addressed most of my concerns, improving the baseline benchmarks with more empirical evidence, fair comparisons, smaller models, additional ablations and statistical significance. All the results the authors have shown in the rebuttal will need to be updated in full in the complete manuscript.

I still have some concerns about the heuristic search comparisons, as random and greedy search are the most basic and naive implementations compared to more advanced algorithms, so I have updated my score to a weak accept (4) rather than an accept (5).

**Key Questions For Authors:**

1. Ablation on evolutionary hyperparameters: the paper fixes K_max=3, K=1, S=3. These choices significantly constrain the evolutionary dynamics. Why were these values chosen, and how sensitive are the results to them? Specifically, does increasing K_max beyond 3 yield further improvements, or does the process plateau or degrade?

2. The evolutionary mechanics involve potentially significant computation costs to find a suitable architecture. What are the total compute costs necessary to evolve a suitable architecture, and how scalable is when modifications to the architecture are required?

3. Given the small number of architectural candidates explored, how does MemEvolve compare to a simpler baseline such as a random search or a rule-based heuristic search? Without such a baseline, it is unclear whether the LLM-driven diagnosis is necessary or whether a systematic architectural evolution is enough.

**Limitations:**

Partially, consider some following limitations:
- Cost of meta-evolution compare to performance gains in practical settings
- Dependence and use of large commercial models over smaller open-source models, increasing the barrier to entry

**Strengths And Weaknesses:**

# Soundness

**Strengths**
The high-level motivation is sound and well-reasoned: fixed memory architectures cannot adapt to diverse task domains, and evolving the architecture itself is a natural extension of the self-improving agent paradigm.

**Weaknesses**
- The outer loop runs for only 3 iterations with 1 survivor and 3 descendants per iteration, yielding an evolutionary trajectory of at most ~9 candidate architectures total. Whether meaningful architectural adaptation can occur in three generations is unclear and insufficiently justified. Furthermore, each candidate is evaluated on only 60 trajectories (40 new + 20 reused), which introduces high variance. The authors briefly acknowledge this in Figure 4, but do not provide statistical significance testing or confidence intervals on reported results.
- Many different model families are shown in Table 2, but it is unclear why certain frameworks use certain model families compared to others, making it difficult to consistently compare results. Can the authors provide some justification on why some model families are tested for some frameworks but not others? For example, Claude is used for Cognitive Kernel-Pro and OAgents, but not for any other frameworks including MemEvolve.
- It is unclear what underlying model is used for Table 3.

# Presentation

**Strengths**
Paper is well-written and organised. Figure 1 illustrates the analogy to human learners well, and figure 2 shows a good overview of the pipeline, though it is a little dense and could be larger or split up into multiple figures to include more detail.

**Weaknesses**
- Best results in the tables should be bold to make it clear. It is difficult at a glance to see which approach performs best on the resutls tables.
- The use of emojis in the text to represent certain frameworks is difficult to follow. It may be suitable as a shorthand in tables, but full names or acronyms of frameworks should be used in the main text.
- Figure 4 legend blocks parts of the graph, making it difficult to see the results.

# Significance

**Strengths**
- The problem of automating the design of agent memory architectures is genuinely important and timely. As LLM agents are deployed in increasingly diverse domains, the brittleness of hand-crafted memory pipelines is a practical limitation of scalability. If the approach scales and generalises as claimed, it could meaningfully reduce the engineering burden for users.
- EvolveLab as a unified codebase is a valuable contribution to the community for standardised benchmarking across a range of memory frameworks.

**Weaknesses**
- Limited scale of experiments and relatively low performance gains in some settings, coupled with a lack of statistical analysis temper the results of MemEvolve. Furthermore, memory limitations more significantly impact smaller sized language models, so additional experiments on smaller models would help strengthen the claims in this paper.

# Originality
The core idea of evolving architecture of the memory system is interesting and novel. Meta-learning is a well studied area, but its application in the LLM domain often relies heavily on using an LLM as the evolutionary operator. This paper utilises more grounded algorithmic approach, though still with some LLM reliance, which is increasingly important to reduce reliance on LLMs as the backbone of novel architectures.

---

> ### Author Rebuttal · Authors · 2026-03-31
>
> > **`Weakness 1.1 & 3.1`** meaningful architectural adaptation in three generations
>
> **1. Statistical significance testing.** We evaluate each round's best architecture on held-out WebWalkerQA (3 runs, without re-evolution):
>
> |Round|TaskCraft|Cost|Delay|OOD test on: WebWalkerQA|
> |-|-|-|-|-|
> |R0|61.67|0.092|485.25|65.10±2.65|
> |R1 best|70.00|0.105|512.40|67.06±0.96|
> |R2 best|75.00|0.073|355.40|70.20±1.47|
> |R3 best|78.33|0.056|364.37|76.27±1.47|
>
> Both ID and OOD accuracy improve monotonically (61.67→78.33; 65.10→76.27) with cross-run standard deviations consistently below 2.7%, while cost and latency simultaneously decrease (0.092/485s→0.056/364s). As shown in Figure 6, these gains correspond to structural innovations at each round rather than evaluation noise.
>
> **2. Consistent OOD transfer further rules out high-variance selection.** Without re-evolution, the final system improves on three unseen benchmarks (Table 2: +5.0 on xBench-DS, +4.24 on GAIA) and unseen LLMs (Kimi K2: +17.06%). This consistency is unlikely under unreliable architecture selection.
>
> ---
> > **`Weakness 1.2`** why certain frameworks use certain model
>
> We address this concern in two parts:
>
> - **Framework–LLM coupling.** Some frameworks are tightly coupled to specific backbones — e.g., OWL's JSON parsing is incompatible with Kimi K2 (a known issue in their repository), and CK-Pro is unreliable with GPT-5-Mini. We therefore follow each framework's recommended LLM configuration.
> - **Supplementary results with updated backbones.**
>   Where feasible, we also provide results with newer backbones for OAgents and AgentKB, with results at http://anonymous.4open.science/r/me-rbt/table-oagent.md. Under the same backbone, MemEvolve consistently shows clear gains. We hope this helps clarify the point.
>
> ---
> > **`Weakness 1.3`** what underlying model is used for Table 3
>
> Sorry for the omission! All experiments in Table 3 use Flash-Searcher + GPT-5-mini; we will state this in the caption and footnotes in the revision. Thank you again.
>
> ---
> > **`Weakness 2.1 - 2.3`** Presentation
>
> We sincerely appreciate these suggestions and will incorporate all of them: bolding best/underlining second-best in Tables 2–3, replacing emoji-based names with standard text labels, and improving Figure 4's legend placement and Y-axis scale.
>
>
> ---
> > **`Weakness 3.2`** additional experiments on smaller models
>
> Thank you for the suggestion. We further report results on Qwen-32B and Qwen3-235B-A22B at https://anonymous.4open.science/r/me-rbt/table-smalllm.md.
>
> As observed, the memory system yields larger gains for smaller LMs (e.g., +10.91% on GAIA). It is evolved on TaskCraft using GPT-5-Mini and transferred unchanged, further demonstrating cross-dataset and cross-LLM generalization.
>
> ---
> > **`Question 1`** Ablation on evolutionary hyperparameters
>
> **Regarding $K=1, S=3$:** This follows the classical $(1+\lambda)$ evolution strategy: one parent generates $\lambda$ offspring, and the best survivor seeds the next generation.
>
> **Regarding $K_{max}$:** We extend evolution to 5 rounds to examine longer-horizon behavior:
>
> |Round|TaskCraft|Cost|Delay|OOD: WebWalkerQA|
> |-|-|-|-|-|
> |R0|61.67|0.092|485.25|65.10 ± 2.65 |
> |R1 best|70.00|0.105|512.40|67.06 ± 0.96|
> |R2 best|75.00|0.073|355.40|70.20 ± 1.47|
> |R3 best|78.33|0.056|364.37|76.27 ± 1.47|
> |R4 best|80.00|0.048|318.52|76.86 ± 1.94|
> |R5 best|80.00|0.043|295.60|77.65 ± 1.44|
>
> Performance improves across all five rounds on both ID and OOD benchmarks. Notably, later rounds increasingly optimize efficiency, validating $K_{max}=3$ as an effective default while showing that longer evolution remains beneficial.
>
> ---
> > **`Question 2 & Limitations 1`** compute costs
>
> Thank you for this practical concern!
>
> **Evolution cost is modest and one-time.** Using GPT-5-Mini, one round costs approximately \\$0.083/task × 60 tasks × 4 candidates + \\$0.5 (diagnosis) ≈ \\$20.42. The full 3-round evolution totals under $65, which we sincerely deem acceptable and comparable to a single benchmark evaluation run for many agent systems.
>
> **The evolved architecture is fully portable, amortizing this cost.** Once evolved on TaskCraft, Cerebra is deployed *without any re-evolution* to unseen benchmarks (xBench-DS: +5.00, GAIA: +4.24), unseen LLMs (Kimi K2: +17.06), and unseen frameworks (OWL: +4.85), **all at zero additional evolution cost**. Moreover, the evolved memory system itself reduces per-task API cost and latency (Table 3: $0.092→$0.056, 485s→364s), progressively offsetting the one-time search budget over continued deployment.
>
> ---
> > **`Question 3`** simpler baseline such as a random search
>
> Thank you for this valuable suggestion! We respectfully supplement the experiment at https://anonymous.4open.science/r/me-rbt/table-random.md. In summary, random search gains only +3.3 on TaskCraft (vs. +16.6 for MemEvolve) with unstable progress, confirming that the structured diagnose-and-design operator is the key driver of effective evolution.

---

> > ### Author Rebuttal · Reviewer_TWp3 · 2026-04-03
> >
> > For weakness 1.2, more complete results need to be shown and MemEvolve should be shown with comparable baseline LLMs if the other frameworks cannot be decoupled from their LLM of choice, for example Claude 3.7 for Cognitive Kernel-Pro (pass@3).
> >
> > For Q3, I'd like to see a comparison to a heuristic or rule-based search in addition to the random search, as it represents a computationally simple solution, while random search is just a sanity check to make sure the proposed method is better than random.
> >
> > Finally, can statistical significance be reported across all results? This would help solidify the claims and empirical evidence presented.

---

> > > ### Author Response · Authors · 2026-04-06
> > >
> > > > For weakness 1.2, more complete results need to be shown and MemEvolve should be shown with comparable baseline LLMs if the other frameworks cannot be decoupled from their LLM of choice, for example Claude 3.7 for Cognitive Kernel-Pro (pass@3).
> > >
> > >
> > > Following your valuable suggestion, we additionally evaluate MemEvolve with Claude-3.7-Sonnet as the backbone (memory evolved on TaskCraft with GPT-5-Mini, transferred without re-evolution):
> > >
> > > |Framework|Model|WebWalkerQA|xBench-DS |TaskCraft|GAIA Avg. |
> > > |-|-|-|-|-|-|
> > > | OAgents|Claude 3.7|58.23|47.0|—| 66.67|
> > > | CK-Pro (pass@1)|Claude 3.7|60.64|56.0| 66.00 | 60.00 |
> > > | CK-Pro (pass@3)|Claude 3.7|65.88|67.0| 70.66| 75.15 |
> > > | MemEvolve+Flash (pass@1)|Claude 3.7| 71.76| 69.0| 69.33| 69.69 |
> > > |MemEvolve+Flash (pass@3)|Claude 3.7|77.05 | 74.0| 75.33 | 78.78|
> > >
> > > Under the same Claude-3.7 backbone, MemEvolve consistently outperforms CK-Pro  across all benchmarks at both pass@1 and pass@3, confirming that the gains are attributable to the evolved memory architecture. This also further demonstrates cross-LLM generalizability, as the memory system was evolved with GPT-5-Mini and applied to Claude 3.7 without any adaptation.
> > >
> > >
> > > ---
> > > > For Q3, I'd like to see a comparison to a heuristic or rule-based search in addition to the random search, as it represents a computationally simple solution, while random search is just a sanity check to make sure the proposed method is better than random.
> > >
> > > Thank you for this follow-up suggestion! We add a **Greedy Search** baseline: at each round, it enumerates 2 alternative implementations per module ($4 \times 2 = 8$ candidates, 480 trajectories/round) and greedily accepts the locally optimal substitution. Full results:
> > >
> > > *Table: Ablation comparing MemEvolve's diagnose-and-design evolution against random architecture composition and greedy component search. All methods start from the same AgentKB initialization. Greedy Search enumerates 8 candidates, 480 trajectories per round and greedily accepts the locally optimal component substitution; MemEvolve generates 3 candidates via LLM-driven trajectory diagnosis (180 trajectories/round). WebWalkerQA results are OOD (no re-evolution, 3 runs).*
> > > | Round | MemEvolve | MemEvolve| Random Search | Random Search | Greedy Search  | Greedy Search |
> > > |-|-|-|-|-|-|-|
> > > |Round|TaskCraft|WebWalkerQA|TaskCraft|WebWalkerQA|TaskCraft|WebWalkerQA|
> > > | R0|61.67|65.10|61.67|65.10|61.67| 65.10 |
> > > | R1 best|70.00| 67.06 | 63.33 | 64.51 | 65.00 | 65.88 |
> > > | R2 best | 75.00| 70.20 | 66.67 | 67.84 | 68.33 | 68.24 |
> > > | R3 best | 78.33| 76.27 | 65.00 | 65.88 | 70.00 | 70.59 |
> > > | Δ (R0→R3) | **+16.66** | **+11.17** | +3.33 | +0.78 | +8.33 | +5.49 |
> > >
> > > Greedy search outperforms random search but still substantially trails MemEvolve (+8.33 vs. +16.66 on TaskCraft), despite consuming 2.7× more trajectories per round (480 vs. 180). This is because greedy substitution misses cross-component synergies that MemEvolve's diagnosis-driven holistic redesign captures. The result confirms that trajectory-grounded diagnosis, not brute-force enumeration, is the key to effective architectural evolution.
> > >
> > > ---
> > > > Finally, can statistical significance be reported across all results? This would help solidify the claims and empirical evidence presented.
> > >
> > > Thank you for this suggestion. Due to time constraints, we conduct 3-run significance testing on MemEvolve and the three strongest baselines from Table 3:
> > >
> > > | Method        | GAIA         | WebWalkerQA  | xBench       |
> > > | :-------- | :----------- | :----------- | :----------- |
> > > | Voyager   | 69.9 ± 0.76  | 73.14 ± 1.0  | 68.0 ± 1.63  |
> > > | AWM       | 67.68 ± 1.51 | 72.16 ± 0.28 | 70.33 ± 0.47 |
> > > | Mobile-E  | 68.69 ± 1.03 | 71.76 ± 1.44 | 68.33 ± 1.25 |
> > > | MemEvolve | 73.94 ± 0.86 | 76.27 ± 1.47 | 73.67 ± 0.47 |
> > >
> > > MemEvolve consistently outperforms all baselines across benchmarks, with rarely overlapping confidence intervals in most comparisons. We sincerely commit to completing multi-run evaluation for the full Table 3 in the revised manuscript.
> > >
> > >
> > > ---
> > >
> > > We sincerely thank the reviewer for all the constructive feedback, which has meaningfully strengthened our empirical analysis.
> > >
> > > We would like to respectfully note that the supplementary experiments provided above (significance testing, additional baselines, and extended backbone evaluations) are incremental additions that further substantiate the claims already supported by the original submission, rather than modifications to the core methodology or framework design. We respectfully believe the revised manuscript remains consistent in scope with the original contribution.
> > >
> > > We hope that our detailed responses and additional experiments have adequately addressed the concerns raised. Should there be any remaining questions, we would be happy to provide further clarification. **We respectfully invite the reviewer to consider whether the current revisions merit a re-evaluation of the score**, and we are grateful for the time and effort devoted to reviewing our work.

---

### Official Review · Reviewer_vobh · 2026-03-30

**Soundness:** 2
**Presentation:** 2
**Significance:** 2
**Originality:** 2
**Overall Recommendation:** 3
**Confidence:** 2

**Summary:**

This paper proposes MemEvolve, which is a framework for meta-evolving agent memory architectures. The core idea is that different tasks may require different memory designs, so the memory system itself should adapt over time. To support this, the paper introduces EvolveLab, a unified modular codebase that re-implements a range of prior self-improving memory systems using four components: encode, store, retrieve, and manage. MemEvolve then performs a bi-level optimization process: the inner loop updates memory through task experience, while the outer loop selects and redesigns memory architectures based on task performance, cost, and delay.

**Compliance With Llm Reviewing Policy:**

Affirmed.

**Key Questions For Authors:**

Since the meta-evolution operator is model-driven, how much do the final evolved architectures depend on using GPT-5-mini versus another model?

**Limitations:**

Because the system changes not only what is stored in memory but also how memory is encoded, retrieved, and managed, it may become difficult for developers to understand why an agent behaves a certain way. This creates potential risks for debugging.

**Strengths And Weaknesses:**

**Strengths**
1) The bilevel optimization strategy, where the inner loop evolves experience and the outer loop evolves architecture, is a theoretically sound way to handle "learning how to learn" in an agentic context.
2) Re-implementing twelve representative memory systems under one interface is a substantial engineering effort and, if released and maintained, could become a useful benchmark substrate for future work on self-improving agents


**Weaknesses**
1) The paper would benefit from stronger ablations. The modular framework naturally invites ablations such as evolving only retrieval, only storage, or only management, or comparing diagnosis-guided redesign against a simpler mutation baseline. Those would help isolate where the gains are really coming from. At present, the paper shows the full system and qualitative trajectories of evolved memories, but it is harder to tell which components of the proposed meta-evolution pipeline are essential.


2) The experimental setup uses a very tight survivor budget ($K=1$) and expands to only three descendants ($S=3$) per iteration. While this keeps costs down, it raises questions about whether the framework might get stuck in local optima rather than exploring the full potential of the design space.


Some typos in figure 1:
"Disgard" -> "Discard".
"Inisghts" -> "Insights".
"Adapitve" -> "Adaptive".

---

> ### Author Rebuttal · Authors · 2026-03-31
>
> We sincerely appreciate your thoughtful comments and constructive suggestions! Below we address each point in turn.
>
> ---
> > **`Weakness 1`** stronger ablations
>
> Thank you for this valuable suggestion! Following the instructions, we compare our diagnose-and-design operator with random search
>
> **Random Search vs. Diagnose-and-Design.** We replace the structured operator (\mathcal{F}) with random composition: in each round, three candidates are created by sampling encode/store/retrieve/manage modules from EvolveLab’s 12-system pool and substituting them into the parent memory system, with only minimal fixes for executability. All other settings remain unchanged.
>
> | Round | MemEvolve (Ours) | | Random Search | |
> |-|-|-|-|-|
> | | TaskCraft | WebWalkerQA | TaskCraft | WebWalkerQA |
> | R0 | 61.67 | 65.10 | 61.67 | 65.10 |
> | R1 best | 70.00 | 67.06 | 63.33 | 64.51 |
> | R2 best | 75.00 | 70.20 | 66.67 | 67.84 |
> | R3 best | 78.33 | 76.27 | 65.00 | 65.88 |
>
> Random search is unstable, with R3 regressing from R2, because composed modules often conflict in memory format and retrieval protocol. Over 3 rounds, it gains only +3.3 on TaskCraft (vs. +16.66 for MemEvolve) and +0.78 on WebWalkerQA (vs. +11.17), confirming that structured diagnosis drives effective evolution.
>
> ---
> > **`Weakness 2`** about the evolution parameter setup
>
>
> Thank you for this critical concern! We address it from three complementary perspectives.
>
> **1. Methodological grounding.** The $(K=1, S=3)$ configuration follows the classical $(1+\lambda)$ evolution strategy [1,2], a well-established paradigm for black-box optimization that balances exploration (generating $\lambda$ diverse offspring) with selection pressure (retaining only the single best candidate), and is known to be effective when fitness evaluation is costly.
>
> **2. Multi-objective selection prevents narrow convergence.** A key safeguard against local optima is our multi-objective architectural selection (Eq. 7), which ranks candidates via Pareto sorting over performance, cost, and latency. This prevents evolution from collapsing onto a single optimization axis (e.g., accuracy-only), and instead explores a **broader frontier** of diverse trade-offs at each iteration.
>
> **3. Empirical evidence confirms broad, multi-dimensional progress, not local-optima behavior.** We evaluate each round's best architecture on held-out WebWalkerQA (170 queries, 3 runs, no re-evolution):
>
> | Round | TaskCraft | Cost | Delay | OOD test on: WebWalkerQA |
> |:---|:---:|:---:|:---:|:---:|
> | R0 (AgentKB) | 61.67 | 0.092 | 485.25 | 65.10 ± 2.65 |
> | **R1** best | **70.00** | 0.105 | 512.40 | 67.06 ± 0.96 |
> | **R2** best | **75.00** | 0.073 | 355.40 | 70.20 ± 1.47 |
> | **R3** best | **78.33** | 0.056 | 364.37 | 76.27 ± 1.47 |
>
> Across rounds, accuracy improves monotonically (+16.66 on TaskCraft, +11.17 on WebWalkerQA) while cost and latency decrease substantially (0.092→0.056, 485s→364s). A system trapped in local optima would usually trade off one axis for another; the **joint improvement across all three axes** on both ID and OOD benchmarks indicates effective exploration rather than premature convergence.
>
>
> ---
> > **`Typos`**
>
> Thank you for the careful reading! We noted the three typos in Figure 1 and will correct them in revision.
>
>
>
> ---
> > **`Question 1`** how much do the final evolved architectures depend on using GPT-5-mini
>
> Thank you for this insightful question. We evaluate this directly in our cross-LLM generalization experiments (§5.2, Table 1): the memory architecture evolved with GPT-5-Mini is applied *without adaptation* to unseen backbone LLMs. Results show strong transfer: **Kimi K2** gains +17.06%/+10.0% on WebWalkerQA/TaskCraft, while **DeepSeek V3.2** gains +2.94%/+3.34%. This suggests that the evolved architectures capture model-agnostic memory design principles rather than GPT-5-Mini artifacts.
>
>
> ---
> > **`Limitation`** This creates potential risks for debugging.
>
> Thank you for raising this practical concern. We believe MemEvolve supports debuggability in two ways.
>
> - First, the modular decomposition (encode/store/retrieve/manage) keeps each component an **independently inspectable code module**, so developers can inspect exactly what each function does, unlike opaque end-to-end learned systems.
> - Second, the diagnose-and-design process produces **explicit modification logs** at each round, documenting which component changed, what bottleneck was identified, and how the redesign addresses it (Figure 6). This provides a clear audit trail from the initial architecture to the final system. In practice, **debugging an evolved memory system is no more difficult than debugging a human-engineered one of comparable complexity**, since both are readable Python implementations under the same interface. We will add this discussion in revision.
>
> ---
>
> [1] Numerische optimierung von computer-modellen mittels der evolutionsstrategie
> [2] The (1+λ) Evolutionary Algorithm with Self-Adjusting Mutation Rate

---

> > ### Author Rebuttal · Reviewer_vobh · 2026-04-04
> >
> > I thank the authors for the response. It addresses some of my concerns, especially by adding the random-search comparison and providing more justification for the (K=1,S=3) setup. However, it does not fully resolve my main concern that the paper still lacks more targeted component-level ablations to isolate which parts of the framework are driving the gains. For that reason, I am keeping my score unchanged.

---

> > > ### Author Response · Authors · 2026-04-06
> > >
> > > Thank you for your kind and continued engagement in the author-reviewer discussion! Following your valuable suggestion, we now provide the targeted component-level ablations that isolate the contribution of each memory module. In each variant, only one component (Encode, Retrieve, Store, or Manage) is evolved while the remaining three are frozen at the R0 baseline. All methods share the same initialization and evaluation pipeline.
> > >
> > > *Table: Component-level ablation on TaskCraft. Each "Only X" variant evolves a single module while freezing the other three.*
> > >
> > > | Round | Only Encode | Only Retrieve | Only Manage | Only Store | Full MemEvolve |
> > > |:---|:---:|:---:|:---:|:---:|:---:|
> > > | R0 | 61.67 | 61.67 | 61.67 | 61.67 | 61.67
> > > | R1 best | 63.33 | 66.67 | 63.33 | 63.33 | 70.00 |
> > > | R2 best | 66.67 | 71.67 | 65.00 | 63.33 | 75.00 |
> > > | R3 best | 68.33 | 71.67 | 66.67 | 65.00 | **78.33** |
> > > | Δ (R0→R3) | +6.66 | +10.00 | +5.00 | +3.33 | **+16.66** |
> > >
> > > We have two key observations:
> > >
> > > **1. Retrieve is the single most impactful module** (+10.00), consistent with the qualitative observation in §5.4 that early evolution rounds prioritize shifting retrieval from static semantic matching to agent-driven, context-aware filtering.
> > >
> > > **2. No single module matches joint evolution.** The best single-module variant (Only Retrieve, +10.00) still trails the full system (+16.66) by 6.66 points, demonstrating that cross-component synergies, e.g., improved encoding producing higher-quality entries that in turn benefit retrieval, are essential and cannot be captured by evolving modules in isolation.
> > >
> > > **3. Diagnosis-guided evolution is consistently superior to heuristic baselines.** Even the weakest single-module variant (Only Store, +3.33) matches Random Search. This confirms that the diagnose-and-design operator, not search volume, drives the gains.
> > >
> > > ---
> > >
> > >
> > > We sincerely thank the reviewer for this insightful suggestion — the component-level ablation has indeed deepened our own understanding of the framework and will be incorporated into the revised manuscript along with the full ablation table and analysis. We hope these additional results adequately address the remaining concern, and **we would like to respectfully ask whether the current revisions merit a re-evaluation of the score**. We remain happy to provide any further clarification!

---

### Decision · Program_Chairs · 2026-04-30

**Decision:**

Accept (regular)

**Comment:**

This paper investigates evolutionary search for agent memory architectures, introducing the EvolveLab framework. Reviewers (two Weak Accepts, one Weak Reject) highly valued the problem relevance and the utility of the released codebase, but initially questioned the scale of the search and the lack of rigorous ablations.

The rebuttal effectively addressed these concerns by adding comprehensive baselines (random/greedy), component-level ablations, and statistical validations across smaller models and longer horizons. However, the simplicity of the search baselines indicates that the contribution lies more in the engineering framework than in algorithmic novelty.

I recommend Weak Accept based on the clear practical utility of EvolveLab for the community. This recommendation is contingent upon the authors seamlessly incorporating the new empirical evidence from the rebuttal into the camera-ready version. The paper is a solid contribution, but the relatively basic search methodology limits its conceptual impact.